# Fine-Grained Captioning of Long Videos through Scene Graph Consolidation

**Sanghyeok Chu** [1]   **Seonguk Seo** [† 1]   **Bohyung Han** [1 2]

## Abstract

Recent advances in vision-language models have led to impressive progress in caption generation for images and short video clips. However, these models remain constrained by their limited temporal receptive fields, making it difficult to produce coherent and comprehensive captions for long videos. While several methods have been proposed to aggregate information across video segments, they often rely on supervised fine-tuning or incur significant computational overhead. To address these challenges, we introduce a novel framework for long video captioning based on graph consolidation. Our approach first generates segment-level captions, corresponding to individual frames or short video intervals, using off-the-shelf visual captioning models. These captions are then parsed into individual scene graphs, which are subsequently consolidated into a unified graph representation that preserves both holistic context and fine-grained details throughout the video. A lightweight graph-to-text decoder then produces the final video-level caption. This framework effectively extends the temporal understanding capabilities of existing models without requiring any additional fine-tuning on long video datasets. Experimental results show that our method significantly outperforms existing LLM-based consolidation approaches, achieving strong zero-shot performance while substantially reducing computational costs.

## 1. Introduction

Vision-language models (VLMs) have demonstrated impressive capabilities across diverse vision-language tasks, including visual question answering, visual dialogue, cross-modal retrieval, and spatiotemporal understanding (Alayrac et al., 2022; Dai et al., 2023; OpenAI, 2023; Chen et al., 2024b; Huang et al., 2024; Zhang et al., 2025; Xu et al., 2024; Maaz et al., 2024). Notably, substantial progress has been made in generating captions for images and short video clips (Liu et al., 2024; Chai et al., 2025; Zhao et al., 2024; Wang et al., 2024; Chen et al., 2024a; Mun et al., 2019).

However, generating captions for longer videos remains a significant challenge. Most existing models are designed for short-term visual inputs, such as images or short video clips, and lack effective support for holistic encoding of entire long videos. As a result, captioning videos beyond a model's temporal window typically requires processing and integrating information from multiple temporal segments. Several approaches, such as memory-based (Zhou et al., 2024; Song et al., 2024; Balazevic et al., 2024) and recursive frameworks (Zhou et al., 2024; Islam et al., 2024; Qian et al., 2024; Weng et al., 2024; Kahatapitiya et al., 2024), have been proposed to consolidate information across these segments. However, these methods often rely on supervised fine-tuning with the target datasets, which limits their generalizability to unseen video domains. More recently, large language models (LLMs) have been employed to generate textual summaries across multiple video segments (Wang et al., 2022b; Chen et al., 2023; Zhang et al., 2024a). While these LLM-based approaches eliminate the need to adapt existing models for long videos, they typically incur high inference overhead and require significant computational resources.

To address these limitations, we propose a novel framework that integrates segment-level captions into a unified global description via graph-based consolidation. We first obtain segment-level captions—each corresponding to either a single frame or a short video clip, depending on the chosen visual captioning model—using an off-the-shelf captioning algorithm. Each caption is then parsed into a scene graph, and these graphs are consolidated into a unified structure that captures the comprehensive semantics of the entire video. Finally, a lightweight graph-to-text decoder, trained solely on external text corpora, translates the consolidated graph into a coherent global caption.

The proposed approach enhances understanding and processing of long-range temporal information without requiring architectural changes or fine-tuning on long video datasets.

[1]ECE, Seoul National University, Korea. [2]IPAI, Seoul National University, Korea. †: Currently at Meta. Correspondence to: Bohyung Han <bhhan@snu.ac.kr>.

*Proceedings of the 42nd International Conference on Machine Learning*, Vancouver, Canada. PMLR 267, 2025. Copyright 2025 by the author(s).

In particular, our framework can be paired with any off-the-shelf VLM, effectively extending its captioning capability beyond the model's inherent temporal constraints. Unlike other LLM-based consolidation methods, it minimizes computational overhead by employing a lightweight graph-to-text decoder with significantly fewer parameters. Our experimental results demonstrate that our approach achieves superior performance in both zero-shot video captioning and zero-shot video paragraph captioning, demonstrating its effectiveness and efficiency.

In summary, our key contributions are organized as follows:

- We propose a novel approach to generate fine-grained captions for long videos using the information across multiple temporal segments.

- We introduce a graph consolidation algorithm that merges segment-level scene graphs into a unified representation to capture both holistic context and fine-grained details across the entire video.

- Our method achieves strong zero-shot captioning performance with significantly lower computational cost compared to LLM-based approaches.

## 2. Related Works

**Video captioning**   Recent advances in video captioning have predominantly rely on supervised training using large-scale datasets, achieving impressive results across various benchmarks (Lei et al., 2021; Wang et al., 2022a; Yan et al., 2022; Liu et al., 2024; Zhao et al., 2024; Wang et al., 2024; Chen et al., 2024a). However, extending these supervised approaches to longer videos remains challenging, primarily due to the scarcity of annotated data covering extensive temporal contexts and the computational complexity involved in modeling long-range dependencies. While various methods have been proposed to tackle these challenges, the needs for supervised fine-tuning for specific target datasets hampers scalability and generalization to unseen video domains (Yang et al., 2023; Islam et al., 2024; Song et al., 2024; Balazevic et al., 2024; Qian et al., 2024; Weng et al., 2024; Kahatapitiya et al., 2024).

**Zero-shot video captioning**   Researchers have explored methods for video captioning without using paired video-text annotations. One approach involves refining language model outputs solely at test time. ZeroCap (Tewel et al., 2022) and related methods (Tewel et al., 2023) use image-text alignment score calculated by CLIP (Radford et al., 2021) in gradient updates to adjust language model features, while MAGIC (Su et al., 2022) employs a CLIP-induced decoding strategy to ensure semantic relevance. Although initially developed for images, these methods extend to videos by aggregating frame-level features into a

single representation. Another approach, often termed zero-shot, involves text-only training without paired video-text annotations, where text decoders are used in conjunction with image-text aligned encoders such as CLIP and Image-Bind (Girdhar et al., 2023). Methods such as DeCap (Li et al., 2023b) and $C^3$ (Zhang et al., 2024b) generate captions by aligning visual and textual features in a shared embedding space. However, these approaches often fail to produce accurate and coherent captions, especially when applied to videos with complex events.

**Zero-shot long video captioning**   Generating coherent and comprehensive captions for long-context videos under zero-shot settings often relies on the consolidation of information derived from multiple temporal segments. Existing consolidation techniques, including memory-based (Zhou et al., 2024; Song et al., 2024; Balazevic et al., 2024) and recursive approaches (Islam et al., 2024; Qian et al., 2024; Weng et al., 2024; Kahatapitiya et al., 2024), require supervised fine-tuning on the target dataset, which limits their applicability to zero-shot scenarios. Recently, LLMs have emerged as a promising tool for zero-shot consolidation, leveraging their general reasoning capabilities without task-specific fine-tuning. For example, VidIL (Wang et al., 2022b) constructs prompts by integrating multi-level textual information from image-language models, including objects, events, attributes, frame captions, and subtitles. Due to the complexity of these prompts, it incorporates illustrative few-shot exemplars from training dataset, to guide LLMs in interpreting and utilizing these textual cues for video captioning Similarly, Video ChatCaptioner (Chen et al., 2023) adopts an interactive framework, where an LLM queries an image VLM for captions of individual frames and aggregates them to generate video caption. While these LLM-based methods are powerful and flexible, they typically incur high computational costs.

## 3. Scene Graph Construction for Videos

To enable effective captioning of long videos, we propose a novel framework that constructs and consolidates scene graphs derived from segment-level captions, as illustrated in Figure 1. The framework comprises four main stages: (1) generating captions for individual video segments using VLMs, (2) converting these captions into scene graphs, (3) merging the scene graphs from all segments into a unified graph, and (4) generating a comprehensive description from the consolidated graph. By aggregating information across segments, the proposed method produces captions that are more coherent and contextually informative, capturing fine-grained details throughout the video. Throughout this paper, we use the term *segment* to denote a temporal unit of a video—either a single frame or a short interval—depending on the characteristics of the employed VLM.

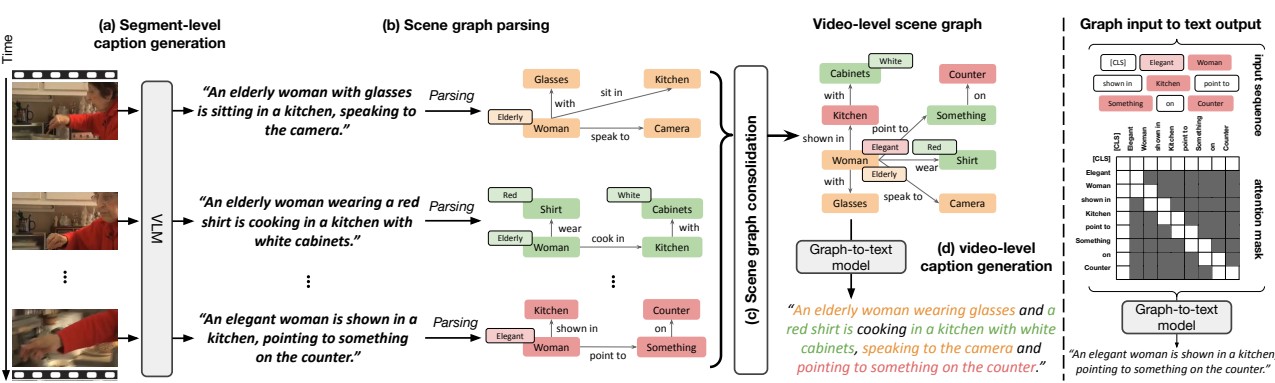

*Figure 1.* An overview of our zero-shot video caption generation framework. (left): The pipeline consists of (a) segment-level caption generation using off-the-shelf VLMs, (b) scene graph parsing for each caption, (c) consolidation of individual scene graphs into a unified graph representing the entire video, and (d) video caption generation through our graph-to-text model. (right): Illustration of how the scene graph is transformed into an input for the graph-to-text model to generate a caption.

## 3.1. Generating segment-level captions

Given an input video, we first divide it into a series of temporal segments. We then generate segment-level captions using off-the-shelf VLMs, with prompts guiding the models to produce descriptive sentences suitable for scene graph construction. While we primarily utilize open-source VLMs as our captioning backbone, our framework is flexible enough to incorporate any VLM, including proprietary or closed-source models, as long as APIs are accessible.

## 3.2. Parsing captions into scene graphs

A scene graph $G = (\mathcal{O}, \mathcal{E})$ is defined by a set of objects $\mathcal{O} = \{o_1, o_2, \ldots\}$, and a set of edges between objects, $\mathcal{E}$. Each object $o_i = (c_i, \mathcal{A}_i)$ consists of an object class $c_i \in \mathcal{C}$ and its attribute set $\mathcal{A}_i \subseteq \mathcal{A}$, where $\mathcal{C}$ is a set of object classes and $\mathcal{A}$ is a set of all possible attributes. A directed edge, $e_{i,j} \equiv (o_i, o_j) \in \mathcal{E}$, has a label $r_{i,j} \in \mathcal{R}$, specifying the relationship from one object to the other. All object classes, attributes, and relationship labels are represented as text strings.

We convert the generated caption from each segment into a scene graph, providing a more structured understanding of each segment. A caption is parsed into a scene graph by textual scene graph parser, and FACTUAL-MR parser (Li et al., 2023c) is used in our implementation. This parser first maps the caption to an intermediate semantic representation consisting of objects, attributes, and relationships, then deterministically converts it into a scene graph. By representing each segment as a graph consisting of objects and their relationships, we can apply a graph merging technique to produce a holistic representation of the entire input video.

## 3.3. Scene graph consolidation

The scene graph consolidation step combines all individual

scene graphs derived from each segment into a unified graph that represents the overall visual content of the video. We first describe our graph merging procedure and then introduce a subgraph extraction technique designed to support more focused and coherent video caption generation.

### 3.3.1. MERGING TWO SCENE GRAPHS

We first describe our scene graph merging technique. Given two scene graphs, $G^s = (\mathcal{O}^s, \mathcal{E}^s)$ and $G^t = (\mathcal{O}^t, \mathcal{E}^t)$, con-

---

**Algorithm 1** Scene graph consolidation

1: **Input:**
2: $\quad \mathcal{G} = \{G_1, G_2, \ldots, G_n\}$: set of scene graphs
3: $\quad \phi(\cdot)$: a graph encoder
4: $\quad \psi_i(\cdot)$: a function returning the $i^{\text{th}}$ object in a graph
5: $\quad \pi$: a permutation function
6: $\quad \tau$: a threshold
7: **Output:** $G_{\text{video}}$: a video-level scene graph
8: **while** $|\mathcal{G}| > 1$ **do**
9: $\quad$ Retrieve the most similar pair $\{G^s, G^t\}$ from $\mathcal{G}$
10: $\quad G^s = (\mathcal{O}^s, \mathcal{E}^s), G^t = (\mathcal{O}^t, \mathcal{E}^t)$
11: $\quad G^m = (\mathcal{O}^m, \mathcal{E}^m) \leftarrow (\mathcal{O}^s \cup \mathcal{O}^t, \mathcal{E}^s \cup \mathcal{E}^t)$
12: $\quad \pi^* \leftarrow \arg\max\limits_{\pi \in \Pi} \sum\limits_i \dfrac{\psi_i(\phi(G^s))}{\|\psi_i(\phi(G^s))\|} \cdot \dfrac{\psi_i(\phi(G^t_\pi))}{\|\psi_i(\phi(G^t_\pi))\|}$
13: $\quad$ **for** $(p, q) \in \mathcal{M}$ such that $s_{p,q} > \tau$ **do**
14: $\quad\quad$ Set the class label of the merged object, $\hat{c}$
15: $\quad\quad \hat{o}_m \leftarrow (\hat{c}, \mathcal{A}^s_p \cup \mathcal{A}^t_q)$
16: $\quad\quad \mathcal{O}^m \leftarrow \{\hat{o}_m\} \cup (\mathcal{O}^m \setminus \{o^s_p, o^t_q\})$
17: $\quad\quad$ Update $\mathcal{E}^m : e_{m,*} \leftarrow e_{p,*}$ and $e_{*,m} \leftarrow e_{*,q}$
18: $\quad$ **end for**
19: $\quad \mathcal{G} \leftarrow \{G^m\} \cup (\mathcal{G} \setminus \{G^s, G^t\})$
20: **end while**
21: $G_{\text{video}} \leftarrow \text{extract}(\mathcal{G})$
22: **return** $G_{\text{video}}$

---

structed from captions corresponding to two different segments, we run the Hungarian algorithm to obtain an optimal matching between the two object sets, $\mathcal{O}^s$ and $\mathcal{O}^t$, which is formally expressed as

$$\pi^* = \arg\max_{\pi \in \Pi} \sum_i \frac{\psi_i(\phi(G^s))}{\|\psi_i(\phi(G^s))\|} \cdot \frac{\psi_i(\phi(G_\pi^t))}{\|\psi_i(\phi(G_\pi^t))\|}, \quad (1)$$

where $\phi(\cdot)$ denotes a graph encoder, $\psi_i(\cdot)$ is a function to extract the $i^{\text{th}}$ object from an embedded graph, and $\pi \in \Pi$ indicates a permutation of objects in a graph. Note that the object matching is based on their cosine similarity, where we introduce dummy objects to deal with different numbers of objects for matching.

After computing all matching pairs using the Hungarian algorithm, we identify a set of valid matches $\mathcal{M}$ by selecting object pairs $(o_p^s, o_q^t)$ whose similarity score $s_{p,q}$ exceeds a predefined threshold $\tau$. For each valid match $(p, q) \in \mathcal{M}$, the merged object $\hat{o}_m \in \hat{\mathcal{O}}$ is defined as

$$\hat{o}_m = (\hat{c}, \mathcal{A}_p^s \cup \mathcal{A}_q^t) \in \hat{\mathcal{O}}, \quad (2)$$

where $\hat{c}$ denotes a class label of a merged object and $\hat{\mathcal{O}}$ represents the set of all merged objects obtained from valid matches. Note that $\hat{c}$ may differ from the original class label of $o_p^s$ or $o_q^t$. This procedure results in a new merged scene graph, $G^m = (\mathcal{O}^m, \mathcal{E}^m)$, which combines each valid pair of matched objects, creating a new object.

We perform graph merging by iteratively selecting and consolidating pairs of graphs based on their embedding similarity. In each iteration, the two most similar graphs are merged into a single graph, which replaces the original pair in the set of graphs. This process is repeated until only one unified scene graph remains. The final scene graph provides a comprehensive representation of the entire video that preserves detailed information from individual segments. Algorithm 1 describes the detailed procedure of our graph consolidation strategy.

### 3.3.2. PRIORITIZED SUBGRAPH EXTRACTION

When concise and focused video captions are desired, we apply subgraph extraction to retain only the most contextually relevant information. During the graph merging process, we track each node's merge count as a measure of its significance within the consolidated graph. We then identify the top $k$ nodes with the highest merge counts and extract their corresponding subgraphs. This approach prioritizes objects that consistently appear across multiple frames, as they often represent key entities in the scene. By focusing on salient elements and filtering out irrelevant details, our method constructs a compact scene graph that enables more focused video captioning.

## 4. Video Caption Generation

Our ultimate goal is to generate captions from a consolidated scene graph. To this end, we develop a graph-to-text decoding model trained on a dataset of graph-text pairs. At inference time, the model takes the consolidated scene graph representing the entire video as input and generates a caption that describes the video as a whole.

### 4.1. Graph-to-text model

Our graph-to-text model consists of a transformer-based graph encoder and a text decoder. The encoder processes the input scene graph to produce a graph embedding, which conditions the decoder to generate the final caption. To reflect the graph topology in our model, we design an attention mask in the graph encoder that restricts attention propagation to the edges defined in the scene graph.

To construct input tokens for the graph encoder, we convert the text values associated with each graph component, such as object classes $c_i$, attribute sets $\mathcal{A}_i$, and edge labels $r_{i,j}$ (*e.g.*, "elderly", "woman", "cook in", "kitchen"), to sequences of embedding vectors. Additionally, we append a learnable embedding token that attends to all other tokens, enabling the aggregation of global context and facilitating information flow across the entire graph, including between disconnected nodes.

### 4.2. Training

We train the graph-to-text model on a large-scale collection of graph-text pairs. To construct this dataset, we curated approximately 2.5 million captions from diverse image captioning datasets, including MS-COCO (Chen et al., 2015), Flickr30k (Young et al., 2014), TextCaps (Sidorov et al., 2020), Visual Genome (Krishna et al., 2017b), and Visual Genome Paragraph Captions (Krause et al., 2017), to cover a broad range of visual scene contexts. To further enrich the dataset, we incorporated model-generated captions for videos in Kinetics-400 (Kay et al., 2017), where LLaVA-NeXT-7B (Liu et al., 2024) is applied to four uniformly sampled frames per video. Each caption is then parsed into a scene graph using a textual scene graph parser, yielding a graph-text pair for training.

Using the graph-text pairs, we train the graph-to-text decoder with a next-token prediction objective, aiming to generate the ground-truth caption conditioned on the input scene graph, as formally defined below:

$$\mathcal{L}(\theta) = \sum_{i=1}^N \log P_\theta(t_i \mid t_{1:i-1}, G), \quad (3)$$

where $t_i$ represents the $i^{\text{th}}$ token in the source text, and $N$ denotes the total number of tokens.

# 5. Experiment

This section presents the effectiveness of the proposed approach through performance evaluation and analysis on both video captioning and video paragraph captioning datasets.

## 5.1. Experimental setup

We provide the detailed information about target tasks with their datasets and baselines. We also discuss a list of performance metrics used in our evaluation.

### 5.1.1. TARGET TASKS AND BASELINES

Our evaluation consists of two zero-shot tasks: (1) video captioning, using the standard test splits of MSR-VTT (Xu et al., 2016) and MSVD (Chen & Dolan, 2011), and (2) video paragraph captioning, using the *ae-val* set of ActivityNet Captions (Krishna et al., 2017a), which contains longer videos with multiple events.

We primarily compare our method against LLM-based approaches. Specifically, we first establish an LLM summarization baseline, which directly summarizes the same set of segment-level captions used by our method. This baseline provides a direct comparison between the proposed scene graph consolidation and the simple aggregation of segment-level captions by LLMs. We use the open-source Mistral-7B-Instruct-v0.3[1] for all datasets. For the ActivityNet Captions dataset, we additionally employ GPT-4o mini, a more powerful proprietary model. Details of the prompt instructions used for the LLM summarization baselines are provided in Appendix B.

We also compare our method against LLM-based video understanding methods, *e.g.*, VidIL (Wang et al., 2022b) and Video ChatCaptioner (Chen et al., 2023), which utilize commercial LLMs along with textual representations derived from VLMs. VidIL constructs rich input sequences by combining various textual cues such as objects, events and frame captions extracted from multiple image-based VLMs, and incorporates few-shot exemplars to guide the LLM in generating video captions. Similarly, Video ChatCaptioner adopts an interactive question-answering framework between image VLM and LLMs.

Note that we primarily focus on LLM-based approaches, as other approaches typically require supervised fine-tuning, making direct zero-shot comparisons infeasible. Additional comparisons with broader zero-shot video captioning approaches—for example, test-time optimization, inference optimization, and text-only training methods—on MSR-VTT are included in the supplementary document.

[1] https://huggingface.co/mistralai/
Mistral-7B-Instruct-v0.3

### 5.1.2. EVALUATION METRICS

Following standard performance evaluation protocols in video captioning, our experiments adopt $n$-gram-based metrics, including BLEU-4 (B@4) (Papineni et al., 2002), METEOR (Banerjee & Lavie, 2005), and CIDEr (Vedantam et al., 2015), which measure the overlap between generated and reference captions. Since these $n$-gram-based metrics are limited in capturing semantic details and contextual accuracy beyond literal phrase matching, we additionally employ BERTScore (Zhang et al., 2020), an embedding-based evaluation metric widely used in natural language processing tasks such as machine translation and summarization. BERTScore measures token-level cosine similarities between generated and reference captions, capturing semantic similarity beyond $n$-gram matches as follows:

$$P_{\text{BERT}} = \frac{1}{|\hat{\mathcal{Z}}|} \sum_{\hat{z}_j \in \hat{\mathcal{Z}}} \max_{z_i \in \mathcal{Z}} z_i^\top \hat{z}_j, \tag{4}$$

$$R_{\text{BERT}} = \frac{1}{|\mathcal{Z}|} \sum_{z_i \in \mathcal{Z}} \max_{\hat{z}_j \in \hat{\mathcal{Z}}} z_i^\top \hat{z}_j, \tag{5}$$

$$F_{\text{BERT}} = \frac{2 \cdot P_{\text{BERT}} \cdot R_{\text{BERT}}}{P_{\text{BERT}} + R_{\text{BERT}}}, \tag{6}$$

where $\mathcal{Z} \equiv \{z_1, z_2, \dots\}$ and $\hat{\mathcal{Z}} \equiv \{\hat{z}_1, \hat{z}_2, \dots\}$ represent the sets of token embeddings in the reference and generated captions, respectively.

## 5.2. Implementation details

Our graph-to-text model consists of a graph encoder and a text decoder, with a total of 235M parameters. The BERT-base model (Devlin et al., 2019) is employed for our encoder, with attention masking as described in Section 4.1, and only the decoder part of T5-base (Raffel et al., 2020) is used as our text decoder.

The graph-to-text model is trained on graph-text pairs constructed in Section 4.2 for $1K$ iterations with a batch size of 512. We employ the AdamW (Loshchilov, 2019) optimizer with a weight decay of 0.05, an initial learning rate of 0.0001, and linear warm-up for the first 1% of training steps. For video paragraph captioning, the model is further fine-tuned for 400 iterations on the subset of the constructed graph-text pairs obtained from the Visual Genome paragraph captioning dataset (Krause et al., 2017).

Segment-level captions are generated using off-the-shelf VLMs. To demonstrate the flexibility of our approach, we employed both image-centric VLMs, including BLIP (Li et al., 2022) and BLIP2 (Li et al., 2023a), and video-centric VLM, InternVL2.5 (Chen et al., 2024a). For MSR-VTT and MSVD, we uniformly sample six frames per video to generate captions using image-centric models. For ActivityNet Captions, we select twelve frames per video when

*Table 1.* Zero-shot video captioning results on the MSR-VTT (Xu et al., 2016) and MSVD (Chen & Dolan, 2011) test sets, comparing our method (SGVC) with LLM-based video understanding methods. † indicates that the method utilizes reference captions from the target dataset to construct few-shot exemplar prompts. Bold numbers indicate the highest scores among methods not using reference captions.

| Dataset | Method | Backbone VLM | B@4 | METEOR | CIDEr | $P_{BERT}$ | $R_{BERT}$ | $F_{BERT}$ |
|---|---|---|---|---|---|---|---|---|
| MSR-VTT | VidIL (Wang et al., 2022b) | BLIP+CLIP | 3.2 | 14.8 | 3.1 | 0.134 | 0.354 | 0.225 |
| | VidIL† (Wang et al., 2022b) | | 13.6 | 20.0 | 20.2 | 0.461 | 0.552 | 0.490 |
| | Video ChatCaptioner (Chen et al., 2023) | BLIP2 | 13.2 | 22.0 | 16.5 | 0.396 | 0.510 | 0.436 |
| | **SGVC (Ours)** | BLIP | 17.7 | 22.5 | 24.0 | **0.476** | 0.539 | **0.490** |
| | | BLIP2 | **18.4** | **23.1** | **26.1** | 0.467 | **0.542** | 0.487 |
| MSVD | VidIL (Wang et al., 2022b) | BLIP+CLIP | 2.5 | 16.5 | 2.3 | 0.124 | 0.404 | 0.238 |
| | VidIL† (Wang et al., 2022b) | | 30.7 | 32.0 | 60.3 | 0.656 | 0.726 | 0.674 |
| | Video ChatCaptioner (Chen et al., 2023) | BLIP2 | 22.7 | 31.8 | 35.8 | 0.496 | 0.651 | 0.550 |
| | **SGVC (Ours)** | BLIP | 22.6 | 30.2 | 50.2 | **0.575** | 0.646 | 0.589 |
| | | BLIP2 | **25.3** | **32.0** | **53.3** | 0.571 | **0.669** | **0.597** |

*Table 2.* Zero-shot video captioning results on the MSR-VTT (Xu et al., 2016) and MSVD (Chen & Dolan, 2011) test sets, comparing SGVC with the LLM summarization baseline. Bold numbers indicate the highest scores.

| Dataset | Method | Backbone VLM | B@4 | METEOR | CIDEr | $P_{BERT}$ | $R_{BERT}$ | $F_{BERT}$ |
|---|---|---|---|---|---|---|---|---|
| MSR-VTT | Summarization w/ Mistral-7B | BLIP | 9.6 | 21.6 | 10.8 | 0.313 | 0.516 | 0.395 |
| | | BLIP2 | 11.5 | **23.1** | 15.4 | 0.308 | 0.528 | 0.397 |
| | **SGVC (Ours)** | BLIP | 17.7 | 22.5 | 24.0 | **0.476** | 0.539 | **0.490** |
| | | BLIP2 | **18.4** | **23.1** | **26.1** | 0.467 | **0.542** | 0.487 |
| MSVD | Summarization w/ Mistral-7B | BLIP | 15.2 | 28.3 | 30.3 | 0.477 | 0.623 | 0.527 |
| | | BLIP2 | 22.5 | 31.9 | 41.6 | 0.500 | 0.664 | 0.558 |
| | **SGVC (Ours)** | BLIP | 22.6 | 30.2 | 50.2 | **0.575** | 0.646 | 0.589 |
| | | BLIP2 | **25.3** | **32.0** | **53.3** | 0.571 | **0.669** | **0.597** |

using image-centric VLMs, while extracting twelve video clips for the video-centric model.

To obtain graph embeddings for Hungarian matching, the graph encoder of our graph-to-text model is used.

For generating the final video caption, we apply a beam search with five beams, a maximum sequence length of 32 and a length penalty of 0.6. For video captioning on MSR-VTT, we apply prioritized subgraph extraction with $k = 1$ to emphasize salient visual information. Video paragraph caption, which requires more detailed descriptions, is generated using a beam search with three beams, a maximum sequence length of 400, and a repetition penalty of 3.0.

### 5.3. Main results

We present quantitative results for zero-shot video captioning on the MSR-VTT and MSVD datasets in Tables 1 and 2, and for zero-shot video paragraph captioning on the ActivityNet Captions *ae-val* set in Tables 3 and 4.

#### 5.3.1. ZERO-SHOT VIDEO CAPTIONING

Table 1 compares the proposed method, SGVC, with existing LLM-based video understanding approaches, VidIL and Video ChatCaptioner. SGVC consistently achieves strong

zero-shot performance across most metrics on both the MSR-VTT and MSVD datasets, outperforming the existing methods. VidIL, although it leverages diverse textual cues from multiple sources, shows limited performance in the zero-shot setting. Notably, SGVC performs competitively even against VidIL's few-shot setting, which heavily depends on dataset-specific exemplars. Video ChatCaptioner, which aggregates information through multi-turn question answering between an LLM and BLIP2, often suffers from hallucinations or overemphasis on irrelevant details, leading to failures in capturing the core content of the video (*e.g.*, "There are no animals present in the park scene.").

Table 2 provides a controlled comparison between SGVC and an LLM-based summarization method, clearly highlighting the effectiveness of our scene graph consolidation approach. Both methods start from an identical set of segment-level captions and this experiments isolates the impact of the graph consolidation. Although LLM summarization produces fluent and expressive captions, it sometimes overlooks details of objects and events within a scene. In contrast, SGVC explicitly integrates segment-level scene graphs into a unified representation, which is helpful for preserving object identities and relationships consistently throughout the video.

*Table 3.* Zero-shot video paragraph captioning results on the ActivityNet Captions (Krishna et al., 2017a) *ae-val* set, comparing our method (SGVC) with LLM-based video understanding methods. † indicates that the method utilizes reference captions from the target dataset to construct few-shot exemplar prompts. Bold numbers indicate the highest scores among methods not using reference captions.

| Method | Backbone VLM | B@4 | METEOR | CIDEr | $P_{BERT}$ | $R_{BERT}$ | $F_{BERT}$ |
|---|---|---|---|---|---|---|---|
| VidIL (Wang et al., 2022b) | BLIP+CLIP | 1.0 | 5.8 | 4.6 | 0.122 | 0.135 | 0.125 |
| VidIL† (Wang et al., 2022b) | | 2.9 | 7.6 | 3.3 | 0.414 | 0.243 | 0.323 |
| Video ChatCaptioner (Chen et al., 2023) | BLIP2 | 2.4 | 8.9 | 1.6 | 0.207 | 0.202 | 0.200 |
| **SGVC (Ours)** | BLIP | 6.7 | 11.6 | 16.6 | **0.367** | 0.285 | 0.322 |
| | BLIP2 | **7.4** | **12.4** | **20.9** | **0.367** | **0.304** | **0.331** |

*Table 4.* Zero-shot video paragraph captioning results on the ActivityNet Captions (Krishna et al., 2017a) *ae-val* set, comparing SGVC with the LLM summarization baselines. Bold numbers indicate the highest scores.

| Method | Backbone VLM | B@4 | METEOR | CIDEr | $P_{BERT}$ | $R_{BERT}$ | $F_{BERT}$ |
|---|---|---|---|---|---|---|---|
| Summarization w/ Mistral-7B | BLIP | 3.4 | 9.4 | 7.5 | 0.292 | 0.268 | 0.276 |
| | BLIP2 | 4.1 | 10.4 | 9.6 | 0.307 | 0.293 | 0.295 |
| | InternVL2.5 | 4.5 | 10.8 | 11.6 | 0.333 | 0.318 | 0.319 |
| Summarization w/ GPT-4o mini | BLIP | 4.6 | 10.2 | 10.3 | 0.325 | 0.284 | 0.300 |
| | BLIP2 | 5.0 | 10.6 | 12.1 | 0.343 | 0.301 | 0.317 |
| | InternVL2.5 | 5.8 | 11.4 | 15.3 | 0.352 | **0.332** | 0.336 |
| **SGVC (Ours)** | BLIP | 6.7 | 11.6 | 16.6 | **0.367** | 0.285 | 0.322 |
| | BLIP2 | 7.4 | 12.4 | 20.9 | **0.367** | 0.304 | 0.331 |
| | InternVL2.5 | **8.0** | **13.2** | **24.1** | 0.359 | 0.326 | **0.338** |

*Table 5.* Comparison of computational costs between SGVC and LLM-based methods on the MSR-VTT test set.

| Method | VLM Backbone | Params. (B) | GPU (GB) | Time (s) | CIDEr | Using reference | Using GPT API |
|---|---|---|---|---|---|---|---|
| VidIL | BLIP+CLIP | 0.67 | 3.57 | 1.32 | 20.2 | ✓ | ✓ |
| Video ChatCaptioner | BLIP2 | 3.75 | 14.53 | 3.65 | 16.5 | - | ✓ |
| Summarization w/ Mistral-7B | BLIP | 7.50 | 14.50 | 1.27 | 10.8 | - | - |
| | BLIP2 | 11.00 | 28.20 | 1.51 | 15.4 | – | – |
| **SGVC (Ours)** | BLIP | 0.74 | 5.07 | 1.14 | 24.0 | - | - |
| | BLIP2 | 4.24 | 18.40 | 1.37 | 26.1 | – | – |

### 5.3.2. ZERO-SHOT VIDEO PARAGRAPH CAPTIONING

Table 3 presents a comparison between SGVC and other LLM-based video understanding methods for zero-shot video paragraph captioning on the ActivityNet Captions *ae-val* set. Consistent with the results observed in zero-shot video captioning in Table 1, SGVC clearly outperforms competing methods. The performance gap is even more pronounced in the paragraph captioning task, where effectively modeling long-range context and maintaining coherence across multiple events is essential.

Table 4 compares SGVC with LLM summarization techniques, using both Mistral-7B and a stronger commercial model, GPT-4o mini. While GPT-4o mini offers significant performance gains over Mistral-7B, it still falls short of SGVC, highlighting the effectiveness of our graph consolidation approach. Furthermore, replacing the backbone captioner with InternVL2.5 further improves SGVC's performance, benefiting from its video-centric design and strong temporal modeling capabilities, despite having significantly fewer parameters than BLIP2 (938M vs. 3.74B). These results clearly demonstrate SGVC's flexibility and plug-and-play compatibility with a wide range of vision-language model architectures.

### 5.4. Analysis

**Efficiency** Table 5 presents a detailed comparison of computational costs, in terms of average per-video inference time and peak GPU memory usage on a single NVIDIA A6000 GPU, along with captioning performance (CIDEr) on the MSR-VTT test set. SGVC consistently outperforms LLM-based summarization approaches across all computational measures, regardless of the underlying backbones. Moreover, our scene graph merging algorithm, which currently runs on the CPU, could be further accelerated by

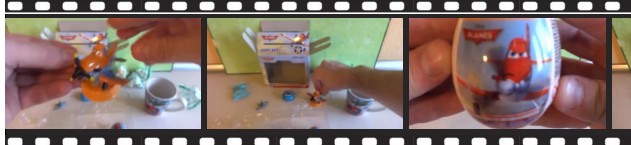

**[Ground-truth]** *A man opening a toy egg set.*

**[LLM summ.]** *A hand opens a toy box, revealing a gift set with a toy airplane, an orange plastic.*

**[VidIL]** *A person is unboxing and demonstrating a toy craft kit.*

**[Video ChatCaptioner]** *The video shows a person holding a toy car and a cup of water while wearing a shirt.*

**[Ours]** *A hand holding a toy airplane in front of a box with a surprised expression.*

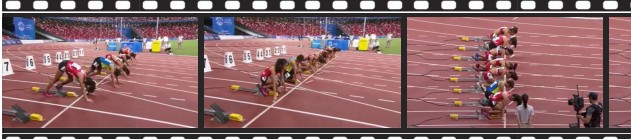

**[Ground-truth]** *A track runner is preparing to run a race.*

**[LLM summ.]** *A group of runners, including females, stretch, crouch at the starting line, and.*

**[VidIL]** *A group of athletes competing in various track and field events.*

**[Video ChatCaptioner]** *The video shows a woman participating in a track and field event, wearing a red shirt and shorts.*

**[Ours]** *A group of runners crouching down a line on a track competing in a race.*

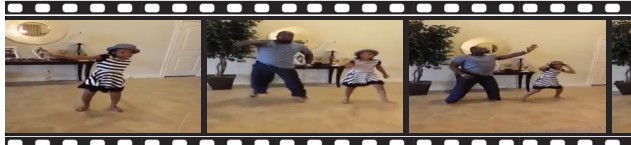

**[Ground-truth]** *A child and a man are dancing to a song.*

**[LLM summ.]** *A family is dancing together in a room, featuring a man, a woman, and a child.*

**[VidIL]** *A father and daughter share a special moment through dance.*

**[Video ChatCaptioner]** *A little girl in a striped dress dances in a living room to an unknown background music.*

**[Ours]** *A man and a young girl dancing in a living room.*

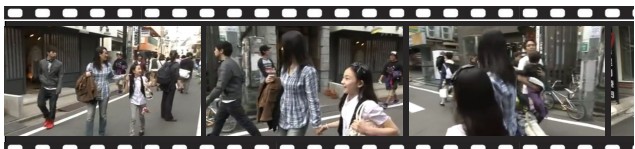

**[Ground-truth]** *A mom and daughter are walking around around town.*

**[LLM summ.]** *A woman and her daughter, accompanied by two other women, are walking down a street.*

**[VidIL]** *A group of people are walking down a street in Japan.*

**[Video ChatCaptioner]** *The video shows a girl wearing a white shirt walking down a street with a bag. The color of the bag is not known.*

**[Ours]** *A woman and her daughter walk down a street with a bicycle in the background.*

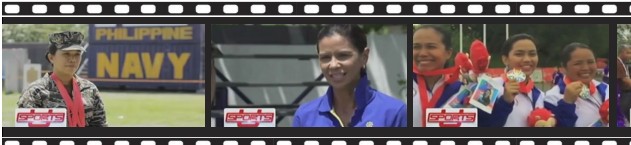

**[Ground-truth]** *A female soldier talks about her athletics.*

**[LLM summ.]** *A woman in a blue jacket poses outdoors, followed by a man in a military uniform standing.*

**[VidIL]** *A female athlete competes in a military-themed sports event.*

**[Video ChatCaptioner]** *The video features a woman in a navy uniform standing in front of a sign that says "phili" with a white wall in the background.*

**[Ours]** *A woman in a uniform stands in front of a sign, holding medals and smiling.*

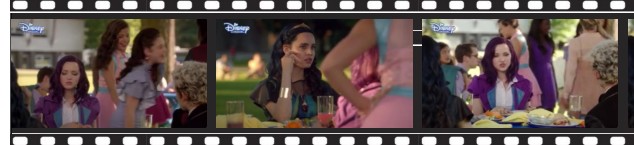

**[Ground-truth]** *People sitting at a table with food.*

**[LLM summ.]** *A television show scene with a man and a woman with long and purple hair, followed by a woman.*

**[VidIL]** *A group of friends are having a dinner party and trying out different hairstyles.*

**[Video ChatCaptioner]** *A group of people are sitting at a table in a park, eating. There are no animals present in the park scene.*

**[Ours]** *A woman in a dress is sitting at a table with food surrounded by people.*

*Figure 2.* Example of zero-shot video captioning results on the MSR-VTT test set. We compare our results with LLM-based methods, listed from top to bottom as 1) LLM summarization using Mistral-7B, 2) VidIL, 3) Video ChatCaptioner, and 4) SGVC (Ours).

*Table 6.* Analysis on the hyperparameter $k$ in the prioritized subgraph extraction, on the MSR-VTT test set.

| $k$ | METEOR | CIDEr | $P_{\text{BERT}}$ | $R_{\text{BERT}}$ | $F_{\text{BERT}}$ |
|---|---|---|---|---|---|
| 1 | 23.1 | **26.1** | **0.467** | 0.542 | **0.487** |
| 3 | **23.8** | 24.9 | 0.454 | **0.554** | 0.486 |

*Table 7.* Analysis on the threshold $\tau$ used in graph consolidation, on the MSVD test set.

| $\tau$ | CIDEr | $F_{\text{BERT}}$ | $\tau$ | CIDEr | $F_{\text{BERT}}$ |
|---|---|---|---|---|---|
| 0.95 | 50.0 | 0.589 | 0.85 | 49.9 | 0.589 |
| 0.90 | 50.2 | 0.589 | 0.80 | 49.9 | 0.589 |

GPU implementation. VidIL and Video ChatCaptioner exhibit slower inference times and lower captioning accuracy. While they consume less GPU memory, their dependence on GPT API calls introduces additional latency.

**Impact of hyperparameters** We analyze the effect of the hyperparameter $k$, which controls the size of the extracted subgraph, as described in Section 3.3.2. As shown in Table 6, lower $k$ values result in more concise subgraphs that emphasize salient objects, leading to improvements in precision-oriented metrics, such as CIDEr and $P_{\text{BERT}}$.

In contrast, higher $k$ values yield richer subgraphs that capture broader contextual information, thereby improving recall-oriented metrics, METEOR and $R_{\text{BERT}}$.

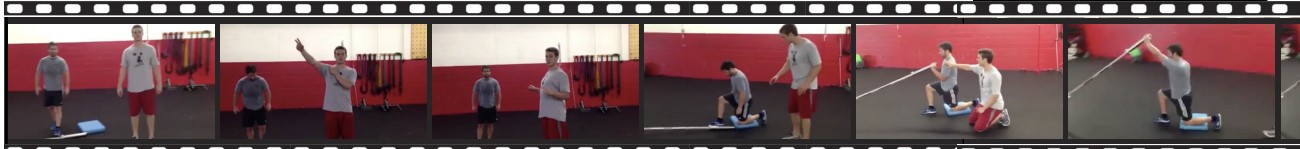

**[Ground-truth]** *Two men are at a gym to demonstrate proper form for the exercise. The man in the black shorts gets on one knee as the instructor gives instructions on what to do. The man in black shorts lifts a bar from the kneeling position. After a few reps, the two men conclude the video.*

**[LLM summ.]** Two men working out in a gym, performing various activities such as weightlifting, martial arts, and stretching.

**[VidIL]** A group of men and women are seen working out in a gym, doing various exercises such as flipping tires, punching bags, and using a mesh sled.

**[Video ChatCaptioner]** The video features a man wearing a black shirt standing on a ledge in front of a red wall indoors. He appears to be leaning forward and looking at the camera with a nervous expression.

**[Ours]** *Two young men are standing in a gym, practicing martial arts. One of the men is holding a baseball. The other man is wearing a gray shirt. The man is standing behind the man. The man is holding a weight. The man is standing with his arms raised.*

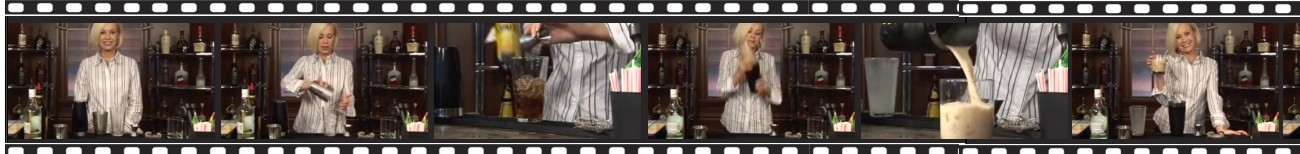

**[Ground-truth]** *A woman pours ice into a glass. She adds shots of alcohol to the glass. She then pours it into another glass and shakes it. She pours that into a glass and sticks a straw in it.*

**[LLM summarization]** *Multiple people are preparing and pouring drinks at a bar, including a woman in a striped shirt.*

**[VidIL]** *A woman is making a drink at a bar.*

**[Video ChatCaptioner]** *The video features a man standing in front of a mirror holding a bottle of wine. In the mirror's reflection, a person and a dog can be seen in a bathroom setting.*

**[Ours]** *A bartender is preparing a drink with a cocktail shaker. She is wearing a striped shirt. The woman is pouring the drink into a glass.*

*Figure 3.* Example of zero-shot video paragraph captioning results on the *ae-val* set of the ActivityNet captions dataset. We compare our results with LLM-based methods, listed from top to bottom as 1) LLM summarization using Mistral-7B, 2) VidIL, 3) Video ChatCaptioner, and 4) SGVC (Ours).

We also conducted evaluation by varying the cosine similarity threshold $\tau$, as reported in Table 7. The results demonstrate stable performance within the range $\tau \in [0.80, 0.95]$, and we set $\tau = 0.9$ for all experiments.

**Qualitative results** Figures 2 and 3 present qualitative examples of zero-shot video captioning on the MSR-VTT test set and video paragraph captioning on ActivityNet Captions *ae-val* set, respectively. Our method generates detailed and contextually rich captions that accurately capture events, objects, and relationships across frames. While LLM summarization and Video ChatCaptioner produce fluent sentences, they occasionally introduce hallucinated content, such as objects or attributes that are not actually present in the video.

## 6. Conclusion

We introduced a novel framework for fine-grained captioning of long videos by consolidating information across multiple temporal segments. Our approach merges scene graphs extracted from segment-level captions to generate comprehensive and coherent video descriptions. This framework provides a computationally efficient and training-free alternative to existing methods. In contrast to LLM-based approaches, our method significantly reduces computational demands by leveraging a lightweight graph-to-text model with substantially fewer parameters. Extensive experiments on both video captioning and video paragraph captioning tasks validate the effectiveness of our method. These results highlight the potential of graph-based consolidation as a foundation for future advances in long video captioning.

## Acknowledgements

We thank Do Young Eun at North Carolina State University for the valuable discussions. This work was supported in part by National Research Foundation of Korea (NRF) grants [RS-2022-NR070855, Trustworthy Artificial Intelligence; RS-2024-00408610, Brain Pool program], Institute of Information & communications Technology Planning & Evaluation (IITP) grants [RS2022-II220959 (No.2022-0-00959), (Part 2) Few-Shot Learning of Causal Inference in Vision and Language for Decision Making; No.RS-2021-II212068, AI Innovation Hub (AI Institute, Seoul National University); No.RS-2021-II211343, Artificial Intelligence Graduate School Program (Seoul National University)] funded by the Korea government (MSIT).

## Impact Statement

The broader impact of this research lies in enabling effective captioning of long videos by leveraging existing vision-language models without any additional fine-tuning on large-scale annotated video datasets. While there is potential for societal impacts arising from this technology, we have not identified any significant negative consequences directly associated with our approach.

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

# A. Additional Experiment

We provide an extended comparison against a broader set of zero-shot video captioning methods on MSR-VTT test set in Table 8.

We compared our approach with several existing approaches, including: 1) test-time optimization via gradient manipulation with CLIP embeddings, *e.g.*, ZeroCap (Tewel et al., 2022) and Tewel *et al.* (Tewel et al., 2023), 2) optimization of inference procedure in the decoder using the CLIP image-text similarity, *e.g.*, MAGIC (Su et al., 2022), and 3) text-only training methods, *e.g.*, DeCap (Li et al., 2023b) and C$^3$ (Zhang et al., 2024b), which are trained solely on text corpora, 4) LLM-based video understanding methods, *e.g.*, VidIL (Wang et al., 2022b) and Video ChatCaptioner (Chen et al., 2023), which utilize proprietary, commercially available LLMs along with textual representations derived from various image-language models, and 5) LLM summarization, which takes the same set of segment-level captions as our method and generates video captions using a pretrained LLM, Mistral-7B-Instruct-v0.3 by text summarization.

Note that DeCap-MSRVTT, C$^3$, and VidIL all utilize annotations from the training dataset but differ in how these annotations are employed. Specifically, DeCap-MSRVTT and C$^3$ use text annotations from the MSR-VTT training set to train their text decoders. In contrast, VidIL constructs few-shot exemplars to serve as prompts, enabling LLM[2] to perform video captioning through in-context learning.

This comprehensive comparison demonstrates that our explicit scene-graph-based modeling achieves superior performance over existing zero-shot video captioning methods across all evaluation metrics.

# B. Prompt Instructions

We provide prompt instructions for segment-level caption generation and LLM summarization of these captions, illustrated here using an image-centric VLM for video captioning.

## B.1. Segment caption generation

Table 9 lists the instructional prompts, generated using GPT-4, which guide VLM in generating the segment-level captions. These prompts are designed to ensure captions remain grounded in the visible content of the image, thereby avoiding factual errors or hallucinated details not supported by the image. A prompt was randomly selected for each segment, allowing captions to reflect diverse aspects of a video.

---

[2]In all our experiments, we use GPT-3.5-turbo-instruct since text-davinci-002 has been deprecated.

## B.2. LLM summarization

To construct the LLM summarization baseline in our experiments, we designed prompts by combining the instructions with segment-level captions, as shown in Table 10. This inputs guide the LLM to generate a concise and coherent video-level summary.

# C. Failure Cases

We present two failure cases from our framework, arising due to hallucinations in the initial segment-level captions.

**Case 1: Incorrect entity counting**

- **Reference captions**: ["A group of people dressed in all of the colors of the rainbow sing a happy song.", "Two elderly women dancing with a group of men.", ... ]
- **SGVC output**: "Two guys in multi-colored tops dance in front of a wall."

While the caption accurately captures specific visual details such as "multi-colored tops", "wall", and "dance", the VLM hallucinate the number of individuals ("two guys", instead of the actual group of "five people").

**Case 2. Object misidentification**

- **Reference captions**: ["A man fixes a piece of machinery that appears to be a miniature tank.", "A guy fixing his camera equipment.", ... ]
- **SGVC output**: "A man is holding a drill in his hand while working on machinery."

The object in the person's hand is a camera, but the initial frame-level captioner incorrectly identified it as a "drill", influenced by the surrounding context. This hallucinated detail was propagated to the final consolidated caption.

# D. Illustration of the Overall Framework

We provide illustrations of the end-to-end flow of our proposed framework for long video captioning, along with additional examples, in Figures 4. The framework includes generating segment-level captions using off-the-shelf VLMs, scene graph parsing for these captions, scene graph consolidation to produce a unified representation, and graph-to-text translation for generate video generation.

# E. Additional Qualitative Results

We provide additional qualitative results for video captioning on the test set of MSR-VTT (Xu et al., 2016) dataset in Figure 5 and for video paragraph captioning on the *ae-val* set of the ActivityNet (Krishna et al., 2017a) Captions dataset in Figure 6.

*Table 8.* Zero-shot video captioning results on the MSR-VTT test set (Xu et al., 2016). ✓ indicates that the method utilizes reference captions from the MSR-VTT dataset. * indicates methods were adapted to zero-shot video captioning by Tewel *et al.* (Tewel et al., 2023). Bold numbers indicate the highest scores among methods not using reference captions.

| Method | Backbone VLM | Using reference | B@4 | METEOR | CIDEr | $P_{\text{BERT}}$ | $R_{\text{BERT}}$ | $F_{\text{BERT}}$ |
|---|---|---|---|---|---|---|---|---|
| *Consolidation-based approaches* | | | | | | | | |
| VidIL (Wang et al., 2022b) | BLIP+CLIP | | 3.2 | 14.8 | 3.1 | 0.134 | 0.354 | 0.225 |
| | | ✓ | 13.6 | 20.0 | 20.2 | 0.461 | 0.552 | 0.490 |
| Video ChatCaptioner (Chen et al., 2023) | BLIP2 | | 13.2 | 22.0 | 16.5 | 0.396 | 0.510 | 0.436 |
| | BLIP | | 9.6 | 21.6 | 10.8 | 0.313 | 0.516 | 0.395 |
| Summ. w/ Mistral-7B | BLIP2 | | 11.5 | 23.1 | 15.4 | 0.308 | 0.528 | 0.397 |
| | LLAVA-Next-7B | | 15.3 | **23.8** | 19.5 | 0.338 | 0.535 | 0.414 |
| | BLIP | | 17.7 | 22.5 | 24.0 | **0.476** | 0.539 | 0.490 |
| **SGVC (Ours)** | BLIP2 | | **18.4** | 23.1 | **26.1** | 0.467 | 0.542 | 0.487 |
| | LLAVA-Next-7B | | 17.1 | 23.0 | 24.0 | 0.455 | **0.547** | **0.497** |
| *Other zero-shot video captioning approaches* | | | | | | | | |
| MAGIC* (Su et al., 2022) | CLIP | | 5.5 | 13.3 | 7.4 | - | - | - |
| ZeroCap* (Tewel et al., 2022) | CLIP | | 2.3 | 12.9 | 5.8 | - | - | - |
| Tewel et al. (Tewel et al., 2023) | | | 3.0 | 14.6 | 11.3 | 0.280 | 0.391 | 0.319 |
| Decap-BookCorpus (Li et al., 2023b) | | | 6.0 | 12.7 | 12.3 | - | - | - |
| Decap-COCO (Li et al., 2023b) | CLIP | | 14.7 | 20.4 | 18.6 | 0.429 | 0.537 | 0.465 |
| Decap-MSRVTT (Li et al., 2023b) | | ✓ | 23.1 | 23.6 | 34.8 | - | - | - |
| $C^3$ (Zhang et al., 2024b) | ImageBind | ✓ | 25.3 | 23.4 | 27.8 | 0.518 | 0.550 | 0.519 |

*Table 9.* The list of instructional prompts for segment-level caption generation using an image-centric VLM.

> • "Please describe what is happening in the image using one simple sentence. Focus only on what is visible."
> • "Now, provide a single sentence caption that describes only what is explicitly shown in the image"
> • "In one sentence, describe what you see in the image without adding any extra details."
> • "Provide a concise one-sentence description of the image, focusing on only the visible elements."
> • "Please give a one-sentence caption that includes only what is clearly shown in the image."
> • "Describe what is happening in the image in one simple sentence, without any added information."
> • "Please generate a single sentence caption that describes only what can be seen in the image."
> • "Provide a one-sentence description of the image, focusing solely on what is shown."
> • "Now, give a brief, one-sentence caption based strictly on the visible content in the image."
> • "In a single sentence, describe what the image shows, without including anything extra."

*Table 10.* Illustration of the input construction for LLM summarization, consisting of the instructional prompt and segment-level captions.

> **Instructional prompt:**
> Below are captions generated from individual frames of a video, each describing specific moments. Please review these frame-by-frame captions and summarize them into a single, compact caption.
>
> **Frame captions:**
> [1 / 6] A woman in a blue jacket is sitting in front of a sports logo.
> [2 / 6] Woman in blue jacket standing outdoors.
> [3 / 6] A man in a military uniform is standing in front of a navy sign.
> [4 / 6] Man in military uniform standing in front of navy sign.
> [5 / 6] The image shows three women wearing sports uniforms and holding medals, smiling and posing for the camera.
> [6 / 6] Three women wearing blue and white uniforms, smiling and holding medals.

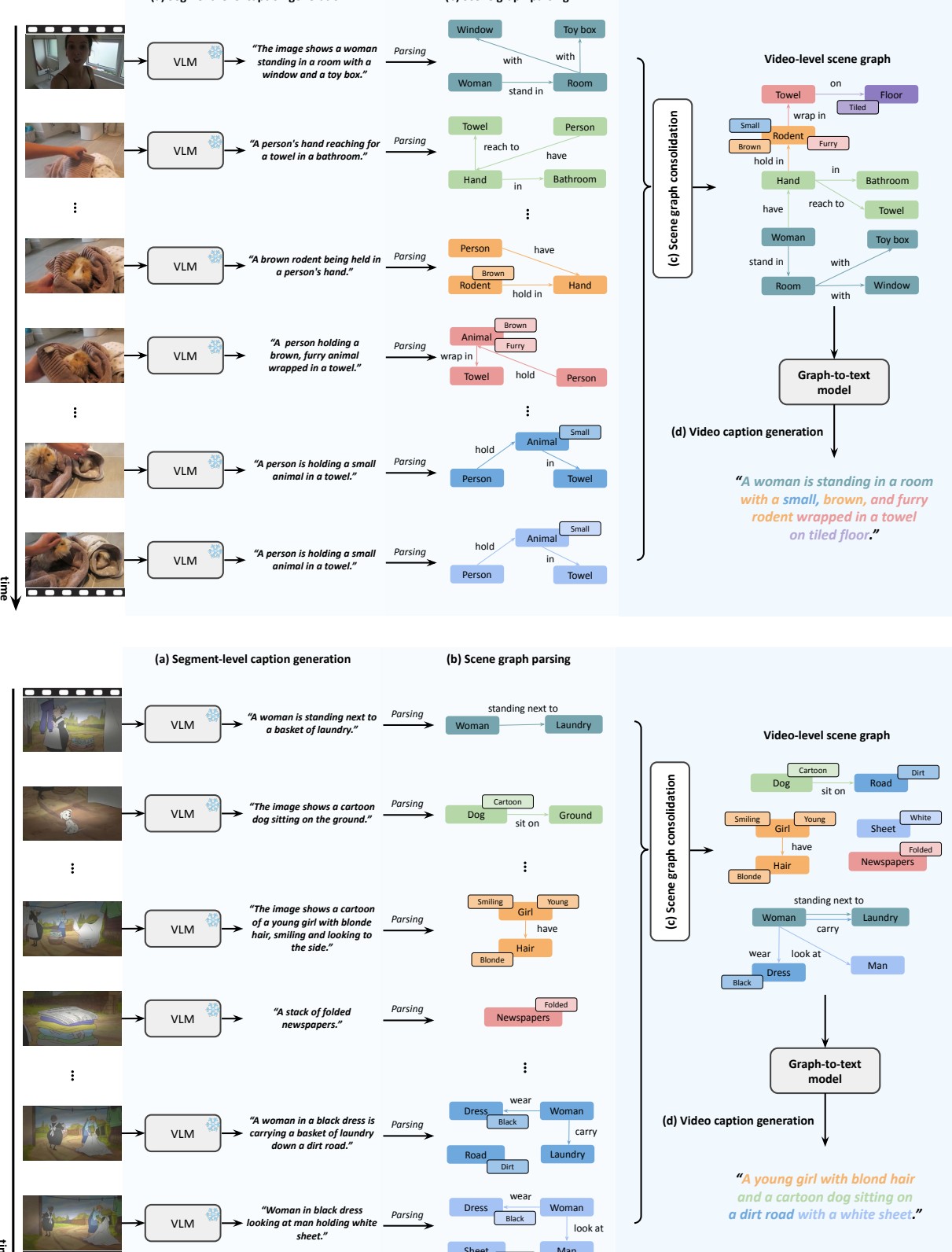

*Figure 4.* Illustrations of the end-to-end flow of the proposed framework. The pipeline consists of: (1) segment-level caption generation via VLMs, (2) scene graph parsing for each segments, (3) scene graph merging to produce a unified representation, and (4) graph-to-text transformation for final caption generation.

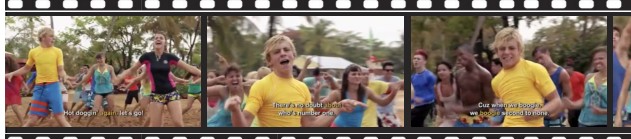

**[Ground-truth]** *A bunch of people dance and sing on a beach.*

**[Tewel et al.]** *Picture shows a caption of the song, which is sung by actor Justin Bieber and features subtitles that read 'I.*

**[Decap-COCO]** *A guy is in the middle of a jump with a frisbee.*

**[C³]** *A group of dancers are dancing in a music video.*

**[LLM summ.]** *A group of people are surfing, dancing, and running on a beach.*

**[VidIL]** *A group of friends are enjoying a day at the beach, surfing and eating hot dogs from a nearby stand.*

**[Video ChatCaptioner]** *A group of people are dancing on the beach to rap music.*

**[Ours]** *A group of people dancing in the sand on a beach with a man in a shirt.*

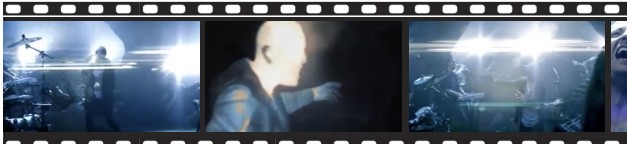

**[Ground-truth]** *A music video for a band showing a crowd of people.*

**[Tewel et al.]** *Picture of the ghostly flame in a song video.*

**[Decap-COCO]** *A person is in the middle of a room with something.*

**[C³]** *A rock band is performing a song.*

**[LLM summ.]** *A person is spotlighted in front of a drum set (1/6), followed by several.*

**[VidIL]** *A band performing in a dark and unconventional setting with a city skyline in the background.*

**[Video ChatCaptioner]** *A group of people are dancing on the beach to rap music.*

**[Ours]** *A group of people playing musical instruments in a dark room with a light shining on their face.*

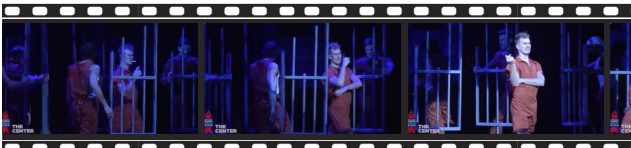

**[Ground-truth]** *A man in a jumpsuit is talking on stage.*

**[Tewel et al.]** *Video shows the actor performing onstage at a musical in New York, which is being held on stage.*

**[Decap-COCO]** *A man is in the middle of a room trying to get on.*

**[C³]** *Tom jones performs on stage.*

**[LLM summ.]** *A man in a red outfit, followed by a man in an orange jumpsuit.*

**[VidIL]** *A theatrical performance with fencing and breakdancing elements.*

**[Video ChatCaptioner]** *The video features a man in an orange jumpsuit standing in front of a cage in a prison setting.*

**[Ours]** *A man in an orange jumpsuit is walking through a gate on stage.*

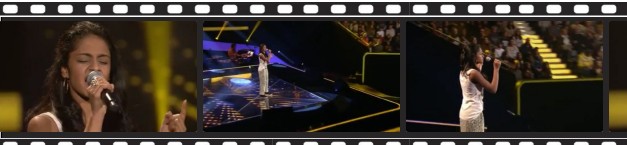

**[Ground-truth]** *A contestant sings in a competition.*

**[Tewel et al.]** *Image shows a contestant singing the song, and another with her hand in it.*

**[Decap-COCO]** *A person is in front of a couple of people on a screen.*

**[C³]** *A girl is performing a song on the voice.*

**[LLM summ.]** *A woman sings on stage, accompanied by a group of young girls and a singer in a red.*

**[VidIL]** *A talent show contestant singing on stage while being judged by a woman in a red chair.*

**[Video ChatCaptioner]** *The video features a woman singing happily into a microphone.*

**[Ours]** *A woman in a white top singing into a microphone on stage.*

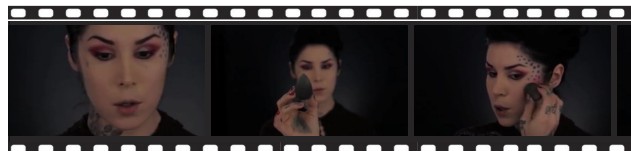

**[Ground-truth]** *A woman with dark makeup is applying more.*

**[Tewel et al.]** *Video showing a makeup artist using her own blood to make an image for the film, 'I'm not a.*

**[Decap-COCO]** *A woman that is holding a phone up to her face.*

**[C³]** *A woman is applying makeup to her face.*

**[LLM summ.]** *A person holds a black object to their face, a woman with tattoos and red nail polish.*

**[VidIL]** *A woman with tattoos and unique style applies makeup in a creative way.*

**[Video ChatCaptioner]** *The video features a woman with a tattoo on her face wearing a black dress in a dark room.*

**[Ours]** *A woman with tattoos applying makeup to her face with a ring.*

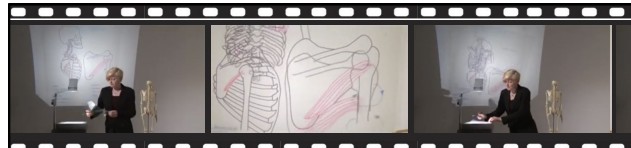

**[Ground-truth]** *A woman gives a presentation on human musculature.*

**[Tewel et al.]** *Picture of anatomy by the author, drawn in animation.*

**[Decap-COCO]** *A person that is in front of a room with a phone.*

**[C³]** *A woman is giving a lecture on a biology equation.*

**[LLM summ.]** *A woman in a black suit is teaching anatomy using a skeleton, anatomical diagram.*

**[VidIL]** *A a physiotherapist is giving a lecture on human anatomy and demonstrating different stretches and exercises for the audience.*

**[Video ChatCaptioner]** *A woman is standing in front of a skeleton. The setting includes a whiteboard with a drawing of a skeleton.*

**[Ours]** *A woman in a suit and black suit is teaching anatomy in front of a projector with a drawing on it.*

*Figure 5.* Additional example of zero-shot video captioning results on MSR-VTT test set. We compare our results with other comparisons, listed from top to bottom as 1) Tewel *et al.*: test-time optimization method, 2) Decap-COCO: text-only training on COCO, 3) C³: text-only training on MSRVTT, 4) LLM summarization using Mistral-7B-Instruct-v0.3, 5) VidIL: LLM-based video understanding with few-shot examples, 6) Video ChatCaptioner: video understanding via multi-turn conversations between VLM and LLM, and 7) SGVC (Ours).

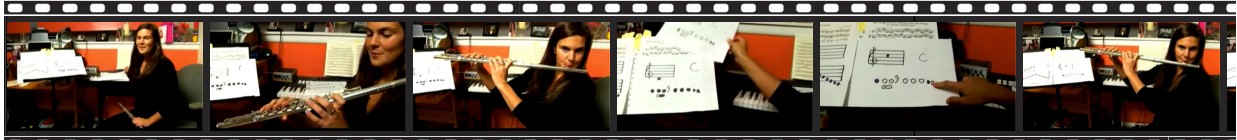

**[Ground-truth]** *We see a lady sitting in front of a keyboard. The lady moves the sheet music. We see the lady shows the keys and pretend to play. We see the lady change the sheet music. We see the lady pretend to play again.*

**[LLM summarizaton]** *A person plays the piano while a woman plays the flute; a music sheet is involved in multiple scenes, and a woman is also seen sitting at a desk with a keyboard.*

**[VidIL]** *A woman is seen playing a silver flute in a music studio.*

**[Video ChatCaptioner]** *The video features a person standing in a living room, wearing a black shirt. The room contains a couch, a chair, and a coffee table. The person is facing the camera and appears to be interacting with objects in the room.*

**[Ours]** *A woman is sitting at a desk in front of a piano playing a flute. The woman is holding a sheet of paper with musical symbols on the stand.*

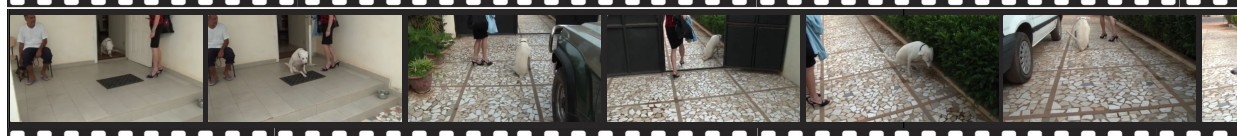

**[Ground-truth]** *A dog is seen walking outside walking runny while one person sits and the other leads the dog. The woman and dog walking around the driveway and the continues walking strange.*

**[LLM summarization]** *A white dog is sitting, standing, and walking on various tiled surfaces, is sniffing bushes and the ground, and is accompanied by a woman in high heels at different points.*

**[VidIL]** *woman is seen walking into a house with a bag in her hand.*

**[Video ChatCaptioner]** *The video features two men interacting in a narrow staircase indoors. The men are not in a hurry and appear to be engaged in physical activity.*

**[Ours]** *A white dog is walking on a tiled walkway in front of a tiled floor. A woman is standing outside next to a woman with a white door. The dog is sniffing a bush near the hedge.*

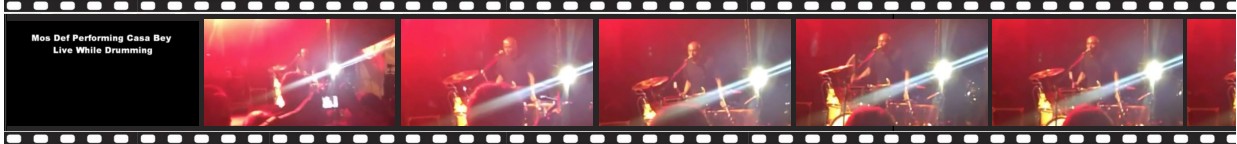

**[Ground-truth]** *A man performs on stage standing and playing a drum set while singing. The crowd waves their hands in the air.*

**[LLM summarization]** *A group of musicians are performing on stage, with a man singing into a microphone, a man playing a guitar, and several men playing drum sets. The stage is illuminated by red lights.*

**[VidIL]** *A band is seen performing on a stage with red lights and a man playing drums.*

**[Video ChatCaptioner]** *The video features a man outdoors hammering a nail into a board using a sledgehammer at a construction site. He is wearing a shirt and shorts, without any safety equipment. The scene has a red and black color scheme.*

**[Ours]** *A man is playing a drum set on stage. The man is singing into a microphone on stage. There is a red light on the stage.*

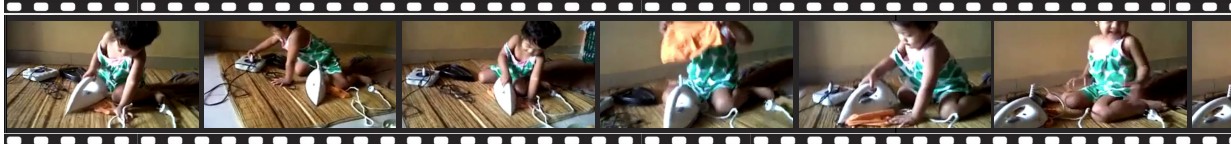

**[Ground-truth]** *A baby sits on a mat on the ground. She plays with an iron. She pretends to iron clothes. She sets the clothes aside.*

**[LLM summarization]** *A young child plays with an iron while various individuals iron clothes on mats or the floor. A baby girl in a green and white dress is also present.*

**[VidIL]** *A child is playing with various household items, including a hair dryer, remote control, and iron.*

**[Video ChatCaptioner]** *The video features a man sitting on a bench in a small garden with a friend. They appear relaxed in the blue and white color scheme of the surroundings.*

**[Ours]** *A young child is sitting on the floor playing with an ironing machine. She is wearing a green dress. The ironing is white. The child is holding an orange cloth.*

*Figure 6.* Additional example of zero-shot video paragraph captioning results on the *ae-val* set of the ActivityNet captions dataset. We compare our results with other comparisons, listed from top to bottom as 1) LLM summarization using Mistral-7B-instruct-v0.3, 2) VidIL, 3) Video ChatCaptioner, and 4) SGVC (Ours).

