# OpenReview forum: "Fine-Grained Captioning of Long Videos through Scene Graph Consolidation"
_ICML.cc/2025/Conference — ICML 2025 poster_

### Official Review · Reviewer_3YCD · 2025-03-12

**Overall Recommendation:** 2

**Summary:**

This paper proposes a zero-shot video captioning framework that leverages scene graph consolidation to bridge image-based vision-language models (VLMs) to video understanding without requiring paired video-text annotations.

**Claims And Evidence:**

Some claims made in the submission are not sufficiently supported.
1. Efficiency Claims: While the authors emphasize "lower inference cost", LLaVA-NEXT-7B’s computational cost (7B parameters) negates this claim compared to smaller VLMs. VideoPrism-B [1] w/ PaLM-1B achieves higher performance with fewer parameters as reported in Table 11 of [2]. Besides, efficiency metrics (FLOPs, latency, or GPU memory usage) are not reported.

[1] Zhao, Long, et al. "Videoprism: A foundational visual encoder for video understanding." arXiv preprint arXiv:2402.13217 (2024).

[2] Wang, Yi, et al. "Internvideo2: Scaling foundation models for multimodal video understanding." European Conference on Computer Vision. Cham: Springer Nature Switzerland, 2024.

**Essential References Not Discussed:**

This paper has discussed enough essential related works.

**Experimental Designs Or Analyses:**

The paper’s experimental design and analysis suffer from critical flaws that undermine the validity of its claims.
1. Misleading Categorization & Incomplete VLM Comparisons: While the authors position their method as "text-only training" (Table 1), the reliance on LLaVA-NEXT-7B (line 305) for frame-level captioning inherently makes this an LLM-based approach. This misleading presentation undermines the fairness of comparisons in Table 1, where LLM-based methods (e.g., Video ChatCaptioner) are treated as separate categories. Besides, the authors fail to directly compare against LLaVA-NEXT-7B itself or state-of-the-art video-centric VLMs like VideoPrism-B [2], which achieves CIDEr=40.3 on MSR-VTT (vs. 24.0 in this work) as reported in Table 11 of [1]. This omission raises concerns about whether the proposed scene graph consolidation adds value over simply using the base VLM’s outputs.
2. Efficiency Claims: While the authors emphasize "lower inference cost," LLaVA-NEXT-7B’s computational cost (7B parameters) negates this claim compared to smaller VLMs. As mentioned above, VideoPrism-B w/ PaLM-1B achieves higher performance with fewer parameters, suggesting the scene graph pipeline introduces bottlenecks rather than enhancements. Besides, efficiency metrics (FLOPs, latency, or GPU memory usage) are not reported.
3. Inadequate Ablation Study: The only ablation study in this paper is about the number of frames, which is not enough to support technical claims. It is necessary to conduct more ablation studies on the VLM backbone (e.g., BLIP-2 vs. LLaVA), hyperparameter k in subgraph extraction, and the graph-to-text model.

[1] Wang, Yi, et al. "Internvideo2: Scaling foundation models for multimodal video understanding." European Conference on Computer Vision. Cham: Springer Nature Switzerland, 2024.

[2] Zhao, Long, et al. "Videoprism: A foundational visual encoder for video understanding." arXiv preprint arXiv:2402.13217 (2024).

**Methods And Evaluation Criteria:**

While the methods align with zero-shot goals, the quantitative experiments are insufficient. Kindly refer to the Experimental Designs Or Analyses section for more details.

**Other Comments Or Suggestions:**

1. Limitations of Image VLMs: The framework relies on LLaVA-NEXT-7B for frame captioning. A discussion of how errors in frame-level captioning propagate to the final video caption (e.g., via case studies) would improve transparency.

**Other Strengths And Weaknesses:**

Strengths:
1. The idea of parsing frame-level captions into scene graphs, then merging them into a unified graph and converting it into a video-level caption is reasonable and interesting.

Weaknesses:
1. Misleading Categorization & Incomplete VLM Comparisons: While the authors position their method as "text-only training" (Table 1), the reliance on LLaVA-NEXT-7B (line 305) for frame-level captioning inherently makes this an LLM-based approach. This misleading presentation undermines the fairness of comparisons in Table 1, where LLM-based methods (e.g., Video ChatCaptioner) are treated as separate categories. Besides, the authors fail to directly compare against LLaVA-NEXT-7B itself or state-of-the-art video-centric MLLMs like VideoPrism-B [2], which achieves CIDEr=40.3 on MSR-VTT (vs. 24.0 in this work) as reported in Table 11 of [1]. This omission raises concerns about whether the proposed scene graph consolidation adds value over simply using the base MLLM’s outputs.
2. Suboptimal Performance: While adopting a multimodal large language model (MLLM) as the backbone, the performance on zero-shot video captioning is inferior compared to other MLLMs and VLMs as reported in [1]. e.g., 24.0 (this work) vs. 40.3 (VideoPrism-B [2] w/ PaLM-1B) in CIDEr scores (↑)  on MSR-VTT.
3. Efficiency Claims: While the authors emphasize "lower inference cost," LLaVA-NEXT-7B’s computational cost (7B parameters) negates this claim compared to smaller VLMs. As mentioned above, VideoPrism-B w/ PaLM-1B achieves higher performance with fewer parameters, suggesting the scene graph pipeline introduces bottlenecks rather than enhancements. Besides, efficiency metrics (FLOPs, latency, or GPU memory usage) are not reported.
4. Inadequate Ablation Study: The only ablation study in this paper is about the number of frames, which is not enough to support technical claims. It is necessary to conduct more ablation studies on the VLM backbone (e.g., BLIP-2 vs. LLaVA), hyperparameter k in subgraph extraction, and the graph-to-text model.
5. Missing Details on Graph-to-Text Model: The technical details on this module are insufficient, restricting reproducibility. It is reasonable to report detailed network architecture. e.g., feature dimensions and how these features are processed using mathematical notations. Besides, the claim of "lightweight" is unsupported without parameter counts or running time metrics versus baselines.

[1] Wang, Yi, et al. "Internvideo2: Scaling foundation models for multimodal video understanding." European Conference on Computer Vision. Cham: Springer Nature Switzerland, 2024.

[2] Zhao, Long, et al. "Videoprism: A foundational visual encoder for video understanding." arXiv preprint arXiv:2402.13217 (2024).

**Questions For Authors:**

1. Performance Gap vs. VideoPrism-B: VideoPrism-VideoPrism-B [1] w/ PaLM-1B CIDEr=40.3 on MSR-VTT, far exceeding your CIDEr=24.0. Does this gap stem from limitations in scene graph consolidation, or is it inherent to using LLaVA-NEXT-7B for frame parsing?
2. Ablation on Subgraph Extraction: How does the choice of k affect caption quality? For instance, does higher k improve coherence at the cost of missing transient objects?

[1] Zhao, Long, et al. "Videoprism: A foundational visual encoder for video understanding." arXiv preprint arXiv:2402.13217 (2024).

**Relation To Broader Scientific Literature:**

The paper’s contributions align with several key directions in vision-language research, particularly in video captioning and multimodal representation learning.

**Theoretical Claims:**

No theoretical claims are made. The paper focuses on empirical validation.

---

> ### Author Rebuttal · Authors · 2025-04-01
>
> **1. Categorization of algorithms**
>
> Given the diversity of zero-shot video captioning approaches, their categorization is not straightforward, and we acknowledge that alternative perspectives exist. However, since SGVC consistently outperforms LLM summarization throughout our experiments, its improved performance is not merely due to the use of LLMs but rather the effectiveness of the proposed algorithmic components. In the revised manuscript, we will reorganize Table 1 to better position SGVC and clarify the sources of its performance gains.
>
>
> **2. Comparison against LLaVA-NEXT-7B itself and VideoPrism-B**
>
> Since LLaVA-NEXT-7B [1] is an image-based captioner, directly evaluating its performance on video captioning is not straightforward. A meaningful approach is to compare LLM summarization and SGVC, as both share the same pipeline for extracting image-level captions using LLaVA-NEXT-7B but differ in their subsequent processing. As shown in Tables 1 and 2, our results clearly demonstrate the superior performance of SGVC over the simple combination of LLaVA-NEXT-7B and LLMs.
>
> VideoPrism-B [2] indeed achieves outstanding performance on MSR-VTT, but it has been trained on an extensive collection of public and private videos with paired captions. As a result, its encoder is specifically optimized for video data, contributing to its superior performance. Additionally, it employs a powerful decoder (PaLM-2) for caption generation, which further enhances its results. Given these factors, we argue that a direct comparison between VideoPrism-B and our method is not entirely fair. Since VideoPrism-B’s model and source code are not publicly available, we instead selected InternVL2.5-1B [3], a video captioner, to assess SGVC’s performance with a video-based model. Our results in Tables D and E demonstrate that SGVC outperforms LLM summarization when integrating a clip-level captioner, highlighting the effectiveness of our graph-based approach.
>
> Table D. Experiments on MSR-VTT, using InternVL2.5-1B as VLM backbone.
>
>
> |Method|B4|METEOR|CIDEr|$P_\text{BERT}$|$R_\text{BERT}$|$F_\text{BERT}$|
> |:-:|:-:|:-:|:-:|:-:|:-:|:-:|
> |LLM summarization|15.1|**23.6**|19.6|0.339|0.532|0.414|
> |SGVC|**17.7**|23.5|**26.7**|**0.481**|**0.550**|**0.513**|
>
>
>
>
> Table E. Experiments on ActivityNet Captions, using InternVL2.5-1B as VLM backbone.
>
>
> |Method|B4|METEOR|CIDEr|$P_\text{BERT}$|$R_\text{BERT}$|$F_\text{BERT}$|
> |:-:|:-:|:-:|:-:|:-:|:-:|:-:|
> |LLM summarization|5.4|11.5|13.0|0.324|0.320|0.322|
> |SGVC|**8.4**|**13.4**|**25.6**|**0.348**|**0.326**|**0.337**|
>
>
> **4. Efficiency**
>
> Please refer to our response to #1 for Reviewer JfkZ.
>
>
> **5. Additional ablation studies**
>
> Please refer to Tables A, B, and C for the ablation studies regarding VLM captioner backbones and hyperparameter $\tau$, respectively.
>
> Additionally, we analyze the impact of the hyperparameter $k$, which determines the size of the extracted subgraph. As shown in Table F, lower $k$ values yield more concise subgraphs centered on salient objects, enhancing precision-oriented metrics (CIDEr, $P_\text{BERT}$). In contrast, higher $k$ values produce richer subgraphs that capture broader contextual information, leading to improvements in recall-oriented metrics (METEOR, $R_\text{BERT}$).
>
> Table F. Ablation study on the hyperparameter $k$, on the MSR-VTT test set.
>
> |$k$|B4|METEOR|CIDEr|$P_\text{BERT}$|$R_\text{BERT}$|$F_\text{BERT}$|
> |:-:|:-:|:-:|:-:|:-:|:-:|:-:|
> |1|17.1|23.0|**24.0**|**0.455**|0.547|**0.497**|
> |3|17.1|**23.7**|22.1|0.428|**0.551**|0.482|
>
>
> **6. Clarification on the graph-to-text model**
>
> Our graph-to-text model consists of a BERT-based encoder and a T5-base decoder, totaling 235M parameters. Given that the other two-stage video captioning models often rely on LLMs, our approach is significantly more lightweight. We will include detailed information about our architecture in the revised version of our paper. More importantly, we plan to release our source code and models upon acceptance of our paper.
>
>
> **7. Error propagation case study**
>
> As mentioned in our response to #3 for Reviewer JfkZ, hallucinations in frame-level caption generation introduce errors in the final video-level captions. We will discuss cases of error propagation in the revised manuscript.
>
>
> [1] Liu, H. et al. LLaVA-NeXT: Improved reasoning, OCR, and world knowledge. https://llava-vl.github.io/blog/2024-01-30-llava-next/, 2024.
>
> [2] Zhao, L. et al. VideoPrism: A Foundational Visual Encoder for Video Understanding. In ICML 2024.
>
> [3] Chen, Z. et al. Expanding Performance Boundaries of Open-Source Multimodal Models with Model, Data, and Test-Time Scaling. In arXiv, 2024.

---

> > ### Comment · Reviewer_3YCD · 2025-04-04
> >
> > I appreciate the authors' response, but my original rating remains unchanged as my concerns about experiment soundness and efficiency reporting (FLOPs, latency, or GPU memory usage) were not addressed.
> >
> > Regarding the authors' missing comparison, Videoprism was one of the papers I referenced. As other reviewer mentioned, this submission actually omitted other relevant works, such as [R1], which achieved a higher CIDEr score (27.1) on MSR-VTT in 2022. Additionally, the experiments were limited to MSR-VTT and ActivityNet, whereas most recent video captioning works also evaluate on Kinetics-400, MSVD, and YouCook2 to demonstrate generalization.
> >
> > [R1] Yan, Shen, et al. "VideoCoCa: Video-text modeling with zero-shot transfer from contrastive captioners." arXiv preprint arXiv:2212.04979 (2022).

---

> > > ### Author Response · Authors · 2025-04-08
> > >
> > > **8. Efficiency report**
> > >
> > > Table G presents a detailed comparison of computational costs, in terms of average per-video inference time and peak GPU memory usage on a single NVIDIA A6000 GPU, alongside captioning performance (CIDEr) on the MSR-VTT test set. Our key findings are summarized below:
> > >
> > > * Decap, which employs a lightweight CLIP backbone, offers faster inference and lower GPU memory usage but achieves a relatively low CIDEr score.
> > > * $C^3$ demonstrates fast inference and strong performance, though its high accuracy may partially stem from its reference to the annotations of the target dataset.
> > > * SGVC consistently outperforms LLM-based summarization approaches across all metrics, regardless of the underlying backbones. Furthermore, our scene graph merging algorithm, which currently runs on the CPU, could be further accelerated by GPU implementation.
> > > * Video ChatCaptioner, VidIL, and [Tewel et al.] are all slower and less accurate. While Video ChatCaptioner and VidIL report lower GPU usage, they introduce additional latency (1+ seconds per API call), which significantly impacts overall efficiency.
> > >
> > > Table G. Computational costs of the compared methods on the MSR-VTT test set. $\alpha$ indicates the latency associated with GPT API calls, which typically take approximately 1+ seconds in our environment.
> > > |Method|VLM backbone|Total Params. (B)|GPU usage (GB)|Inference time (s)|CIDEr|Using ref.|Using GPT API|
> > > |:-:|:-:|:-:|:-:|:-:|:-:|:-:|:-:|
> > > |Decap|CLIP|0.22|2.87|0.17|18.6|-|-|
> > > |C^3|ImageBind|1.33|5.28|0.23|27.8|$\checkmark$|-|
> > > |LLM summarization|BLIP|7.50|14.50|1.27|10.8|-|-|
> > > |**SGVC**|BLIP|0.74|5.07|1.14|24.6|-|-|
> > > |LLM summarization|BLIP2|11.00|28.20|1.51|15.4|-|-|
> > > |**SGVC**|BLIP2|4.24|18.40|1.37|26.9|-|-|
> > > |Video ChatCaptioner|BLIP2|3.75|14.53|(0.10+$\alpha$) $\times$ 30|16.5|-|$\checkmark$|
> > > |VidIL|BLIP+CLIP|0.67|3.57|0.20+$\alpha$|19.4|$\checkmark$|$\checkmark$|
> > > |Tewel et al.|CLIP|0.51|2.40|83.30|11.3|-|-|
> > >
> > > **9. Comparisons with VideoPrism and VideoCoCa**
> > >
> > > Although VideoPrism and VideoCoCa are relevant, a fair comparison with our approach (SGVC) is difficult due to differences in algorithm design and evaluation protocols.
> > >
> > > In addition to the discussion in #2 of our previous response, VideoPrism is specifically designed and trained for short video clips. As such, holistic encoding of an entire long video is not well-supported by the model. To apply VideoPrism to long video captioning, an additional module would be required to overcome this limitation—yet no such extension is proposed in the original paper. Furthermore, VideoPrism does not report performance on the ActivityNet Captions benchmark. While results on YouCookII are provided, they are based on individual segments rather than full videos.
> > >
> > > VideoCoCa adopts a pipeline more similar to ours, utilizing attentional pooling to consolidate features from multiple frames. While VideoCoCa slightly outperforms SGVC on MSR-VTT (as shown in Table A of our rebuttal to Reviewer JfkZ), SGVC demonstrates stronger performance on the long-video dataset, although we admit that the comparison is not entirely fair. ActivityNet Captions, as presented in Table B of our rebuttal to Reviewer JfkZ. The experimental protocol used for YouCookII is identical to that of VideoPrism, which means VideoCoCa has not been properly evaluated on this dataset.
> > >
> > > We are eager to conduct comprehensive comparisons with both VideoPrism and VideoCoCa using reproduced results. Unfortunately, this is not feasible, as the authors have not released their models or pretrained weights. As such, direct comparisons based on the numbers reported in their papers may be misleading. We kindly ask reviewers to take this context into account when evaluating our work.
> > >
> > > **10. Further ablation study on LLM summarization**
> > >
> > > We further validated SGVC by comparing it to LLM summarization with a stronger LLM, GPT-4o-mini. SGVC generally outperforms GPT-4o-mini across all VLM backbones, confirming the advantage of graph-based consolidation.
> > >
> > > Table H. Ablation study of LLM summarization on ActivityNet Captions val set. LLM (M), LLM (G) denotes LLM summarization using Mistral-7B and GPT-4o-mini, respectively. "*" denotes the same backbone VLM as the row above.
> > > |Method|Backbone VLM|B4|METEOR|CIDEr|$P_\text{BERT}$|$R_\text{BERT}$|$F_\text{BERT}$|
> > > |:-:|:-:|:-:|:-:|:-:|:-:|:-:|:-:|
> > > |LLM (M)|BLIP|4.0|10.1|8.7|0.304|0.277|0.290|
> > > |LLM (G)|*|4.6|10.3|10.4|0.324|0.285|0.303|
> > > |SGVC|*|**7.1**|**11.9**|**17.5**|**0.366**|**0.288**|**0.322**|
> > > |LLM (M)|BLIP2|4.8|11.0|10.5|0.319|0.300|0.309|
> > > |LLM (G)|*|5.0|10.5|12.8|0.343|0.300|0.320|
> > > |SGVC|*|**7.9**|**12.9**|**22.6**|**0.369**|**0.310**|**0.337**|
> > > |LLM (M)|LLAVA-Next-7B|4.7|11.0|10.3|0.297|0.303|0.300|
> > > |LLM (G)|*|4.7|10.5|12.9|0.311|0.299|0.305|
> > > |SGVC|*|**7.4**|**12.5**|**22.0**|**0.342**|**0.308**|**0.324**|
> > > |LLM (M)|InternVL2.5-1B|5.4|11.5|13.0|0.324|0.320|0.322|
> > > |LLM (G)|*|5.8|11.4|16.0|0.330|**0.328**|0.329|
> > > |SGVC|*|**8.4**|**13.4**|**25.6**|**0.348**|0.326|**0.337**|

---

### Official Review · Reviewer_JfkZ · 2025-03-13

**Overall Recommendation:** 3

**Summary:**

This paper proposes a zero-shot video captioning method utilizing frame-level scene graphs obtained from image-based Visual-Language Models (VLMs). The authors consolidate these frame-level graphs into a unified video-level representation using Hungarian matching, followed by generating video-level captions through a lightweight graph-to-text decoder trained only on text corpora. Experimental evaluations on standard benchmarks (MSR-VTT, ActivityNet Captions) demonstrate superior performance compared to existing zero-shot methods, approaching supervised results.

**Claims And Evidence:**

Most claims are supported with clear empirical evidence, such as thorough experimental results using multiple established metrics (BLEU, METEOR, CIDEr, BERTScore). However, the claim regarding inference-time efficiency is not sufficiently supported by explicit timing comparisons with baseline methods. The claim that the approach reduces hallucinations is qualitative rather than quantitatively supported. Quantitative evaluation would strengthen this claim.

**Essential References Not Discussed:**

The key contribution of the paper is the generation and consolidation of Vision-Language Model (VLM) captions to enhance zero-shot video captioning. The proposed approach effectively addresses the problem with a well-structured process. There are no critical missing references.

**Experimental Designs Or Analyses:**

Almost of the experimental designs make sound. However, it is difficult to visually recognize the results of Table 1 with and without reference captions because they are directly attached to the SGVC.

**Methods And Evaluation Criteria:**

The proposed methods are well-reasoned and effectively address the zero-shot captioning challenge. The benchmarks and evaluation metrics are also appropriate.

**Other Comments Or Suggestions:**

N/A

**Other Strengths And Weaknesses:**

__Strength__

S1. Novelty in applying scene graph consolidation for zero-shot video captioning.

S2. Clear methodological explanations supported by strong experimental validation.

S3. Effectively addresses practical limitations inherent in zero-shot captioning scenarios.

__Weakness__

W1. Lack of evaluation regarding inference-time cost.

W2. Missing ablation studies examining the impact of different Visual-Language Models.

W3. Can the authors provide the case of captioning failure? (e.g. due to VLM hallucination)Suggestion 1. Can you provide the case of captioning failure? (e.g. due to VLM hallucination)

W4. How sensitive is the performance of your method to varying the threshold (τ) during graph merging?

W5. What are the criteria for selecting frames in an ablation study for the number of frames in Table 3?

**Questions For Authors:**

See the weakness.

**Relation To Broader Scientific Literature:**

The paper situates its contributions within the current literature well, citing recent works on zero-shot video captioning and Dense Video Captioning (DVC) frameworks. Notably, it introduces a novel framework that leverages scene graphs, a method not commonly used in previous zero-shot video captioning research.

**Theoretical Claims:**

The paper makes no explicit theoretical claims. It is primarily empirical and methodological. The proposed methods are practical, with clear descriptions and validation through empirical studies.

---

> ### Author Rebuttal · Authors · 2025-04-01
>
> **1. Inference cost**
>
> SGVC is efficient in terms of latency, comparable to Decap and C3. However, direct comparisons are challenging as each method utilizes different backbone models for image captioning (SGVC: BLIP, BLIP2, LLAVA-NEXT-7B; Decap: CLIP; C3: ImageBind). In contrast, Video ChatCaptioner and [Tewel et al.] are significantly slower, by at least an order of magnitude, due to the former requiring multiple evaluation rounds and the latter involving test-time optimization.
>
> For the performance of SGVC with BLIP and BLIP2, please refer to our response for Tables A and B in #2. These models actually outperform our implementation with LLAVA-NEXT-7B.
>
> **2. Different VLM captioning models**
>
> We evaluate our approach using various VLM backbones, including image-based models (BLIP, BLIP-2, LLaVA-NEXT-7B) and a video-based model (InternVL2.5-1B [1]), on the MSR-VTT (Table A) and ActivityNet Captions (Table B) datasets. The results consistently demonstrate that SGVC outperforms its LLM summarization counterpart, regardless of the chosen VLM backbone. Note that the SGVC results with LLaVA-NEXT-7B in Table B differ slightly from those in the original manuscript, as we extended fine-tuning on Visual Genome paragraphs from 500 to 3200 iterations.
>
> Table A. Ablation of VLM backbones on MSR-VTT test set.
> |Method|Backbone VLM|Using ref.|B4|METEOR|CIDEr|$P_\text{BERT}$|$R_\text{BERT}$|$F_\text{BERT}$|
> |:-:|:-:|:-:|:-:|:-:|:-:|:-:|:-:|:-:|
> |VidIL|BLIP+CLIP|$\checkmark$|13.3|20.3|19.4|0.452|**0.553**|0.486|
> |LLM summarization|BLIP||9.6|21.6|10.8|0.313|0.516|0.390|
> |SGVC|BLIP||**17.9**|**22.4**|**24.6**|**0.485**|0.536|**0.509**|
> |Video ChatCaptioner|BLIP2||13.2|22.0|16.5|0.396|0.510|0.436|
> |LLM summarization|BLIP2||11.5|**23.1**|15.4|0.308|0.528|0.389|
> |SGVC|BLIP2||**18.6**|23.0|**26.9**|**0.475**|**0.540**|**0.505**|
> |LLM summarization|LLAVA-Next-7B||15.3|**23.8**|19.5|0.338|0.535|0.414|
> |SGVC|LLAVA-Next-7B||**17.1**|23.0|**24.0**|**0.455**|**0.547**|**0.497**|
> |LLM summarization|InternVL2.5-1B||15.1|**23.6**|19.6|0.339|0.532|0.414|
> |SGVC|InternVL2.5-1B||**17.7**|23.5|**26.7**|**0.481**|**0.550**|**0.513**|
>
> Table B. Ablation of VLM backbones on ActivityNet Captions val set.
> |Method|Backbone VLM|B4|METEOR|CIDEr|$P_\text{BERT}$|$R_\text{BERT}$|$F_\text{BERT}$|
> |:-:|:-:|:-:|:-:|:-:|:-:|:-:|:-:|
> |LLM summarization|LLAVA-Next-7B|4.7|11.0|10.3|0.297|0.303|0.300|
> |SGVC|LLAVA-Next-7B|**7.4**|**12.5**|**22.0**|**0.342**|**0.308**|**0.324**|
> |LLM summarization|BLIP|4.0|10.1|8.7|0.304|0.277|0.290|
> |SGVC|BLIP|**7.1**|**11.9**|**17.5**|**0.366**|**0.288**|**0.322**|
> |LLM summarization|BLIP2|4.8|11.0|10.5|0.319|0.300|0.309|
> |SGVC|BLIP2|**7.9**|**12.9**|**22.6**|**0.369**|**0.310**|**0.337**|
> |LLM summarization|InternVL2.5-1B|5.4|11.5|13.0|0.324|0.320|0.322|
> |SGVC|InternVL2.5-1B|**8.4**|**13.4**|**25.6**|**0.348**|**0.326**|**0.337**|
>
> **3. Failure cases**
>
> We provide two failure cases given by hallucinations in the initial image-based caption:
>
> Example 1.
> * Reference captions: ["A group of people dressed in all of the colors of the rainbow sing a happy song.", "Two elderly women dancing with a group of men.", …]
> * SGVC output: "Two guys in multi-colored tops dance in front of a wall."
>
> While the caption accurately captures specific visual details such as "multi-colored tops", "wall", and "dance", the VLM hallucinates the number of individuals (reporting "two guys" instead of the actual group of five people).
>
> Example 2.
> * Reference captions: ["A man fixes a piece of machinery that appears to be a miniature tank.", "A guy fixing his camera equipment.", … ]
> * SGVC output: "A man is holding a drill in his hand while working on machinery."
>
> The object in the person's hand is a camera, but the initial frame-level captioner incorrectly identified it as a "drill", influenced by the surrounding context. This hallucinated detail was propagated to the final consolidated caption.
>
> **4. Impact of threshold $\tau$ in graph merging**
>
> We conducted an ablation study by varying the cosine similarity threshold, $\tau$, as shown in Table C. The results demonstrate stable performance within the range $\tau \in [0.80, 0.95]$.
>
> Table C. Ablation study on the threshold $\tau$.
> |$\tau$|B4|METEOR|CIDEr|$P_\text{BERT}$|$R_\text{BERT}$|$F_\text{BERT}$|
> |:-:|:-:|:-:|:-:|:-:|:-:|:-:|
> |0.95|17.1|23.0|24.0|0.455|0.547|0.497|
> |0.90|17.3|22.5|24.5|0.463|0.542|0.499|
> |0.85|17.2|22.8|23.8|0.455|0.545|0.496|
> |0.80|16.9|22.8|23.0|0.445|0.546|0.490|
>
> **5. Frame selection criteria**
>
> In our experiment for Table 3, we uniformly sampled frames to ensure coverage of the entire video content.
>
> **6. Quantitative evaluation of hallucination reduction**
>
> Thank you for your suggestion. However, conducting a quantitative evaluation within a short time frame is challenging due to the lack of necessary annotations and evaluation metrics.
>
> [1] Chen, Z. et al. Expanding Performance Boundaries of Open-Source Multimodal Models with Model, Data, and Test-Time Scaling. In arXiv 2024.

---

### Official Review · Reviewer_38AV · 2025-03-13

**Overall Recommendation:** 3

**Summary:**

This paper introduces a novel zero-shot video captioning approach that leverages scene graphs to bridge image and video understanding without requiring paired video-text data. The four-step process involves: generating frame-level captions using an image VLM, converting these into scene graphs, consolidating the graphs into a unified video-level representation through a novel merging algorithm, and finally generating captions using a lightweight graph-to-text model trained only on text corpora.Evaluated on MSR-VTT and ActivityNet Captions datasets, the method outperforms existing zero-shot baselines while reducing computational costs.

**Claims And Evidence:**

Yes.

**Essential References Not Discussed:**

Some important methods are not discussed, such as [1, 2, 3], and some important benchmarks were not used for evaluation, such as [4].

[1] Huang, B., Wang, X., Chen, H., Song, Z., & Zhu, W. (2024). Vtimellm: Empower llm to grasp video moments. In Proceedings of the IEEE/CVF Conference on Computer Vision and Pattern Recognition (pp. 14271-14280).

[2] Tang, Y., Shimada, D., Bi, J., Feng, M., Hua, H., & Xu, C. (2024). Empowering LLMs with Pseudo-Untrimmed Videos for Audio-Visual Temporal Understanding. arXiv preprint arXiv:2403.16276.

[3] Kim, M., Kim, H. B., Moon, J., Choi, J., & Kim, S. T. (2024). HiCM2^2
2: Hierarchical Compact Memory Modeling for Dense Video Captioning. arXiv preprint arXiv:2412.14585.

[4] Zhou, L., Xu, C., & Corso, J. (2018, April). Towards automatic learning of procedures from web instructional videos. In Proceedings of the AAAI conference on artificial intelligence (Vol. 32, No. 1).

Please also see the "Other Strengths And Weaknesses" part.

**Experimental Designs Or Analyses:**

The experimental design is sound overall, with appropriate datasets, baselines, metrics.

**Methods And Evaluation Criteria:**

Yes, the proposed methods make sense to me.

**Other Comments Or Suggestions:**

N/A

**Other Strengths And Weaknesses:**

Strengths:
S1: The approach cleverly addresses the scarcity of video-text paired data by leveraging existing image VLMs and text-only training.
S2: The scene graph consolidation represents an elegant solution to maintain consistency in object identity across frames, addressing a common issue in naive frame-based approaches.

Weaknesses:
W1: The approach relies on the quality of the initial frame-level captions and scene graph parsing, creating potential error propagation throughout the pipeline.
W2: The performance of 'LLM summarization' on ANet Caps shown in Table 2 seems to be too low. Some LLM-based methods mentioned in "Essential References Not Discussed" part can achieve better performance than that.

**Questions For Authors:**

Q1: How robust is the scene graph consolidation algorithm to errors in the initial scene graph parsing? Since the method relies on accurate object detection and relationship parsing, I'm curious whether you've analyzed how errors in this initial stage propagate through the pipeline.

Q2: The ActivityNet videos are relatively short compared to many real-world video understanding scenarios. Do you consider evaluating your method on benchmarks with longer videos, such as the [4] mentioned in "Essential References Not Discussed" part?

**Relation To Broader Scientific Literature:**

The use of scene graphs as structured intermediate representations connects to prior work on scene graph generation and reasoning in the image domain. The graph consolidation approach extends ideas from temporal video understanding while addressing the data scarcity problem in video captioning through a zero-shot framework.

The work also relates to recent efforts bridging image and video domains using large language models and creating more structured representations for video understanding beyond simple frame averaging.

**Theoretical Claims:**

This paper doesn't contain formal mathematical proofs or theoretical claims requiring verification.

---

> ### Author Rebuttal · Authors · 2025-04-01
>
> **1. Reliance on the quality of frame-level captions and vulnerability to cumulative parsing errors in scene graphs**
>
>
> *  Error propagation is common in other algorithms:
> The primary objective of this work is to understand video content and generate fine-grained, video-level captions without encoding entire videos at once. This approach is especially effective for handling long videos, which require division into multiple segments or frames. However, consolidating information from these segments inevitably leads to cumulative errors—a challenge common to various algorithms despite their efforts to minimize it.
>
> *  Error quantification is difficult:
> Due to the zero-shot nature of our method, directly quantifying the cumulative error from abstracting segment-level (or frame-level) visual information into textual captions and extracting video-level representation based on a scene graph generation and consolidation is challenging. This is because existing methods employ fundamentally different algorithms, including variations in encoder and decoder choices for representation and captioning.
>
> *  An attempt to estimate the impact of cumulative parsing error:
> One way to isolate the parsing error in scene graph construction is to compare performance between LLM summarization and our approach, SGVC. Note that LLM summarization follows the same pipeline as SGVC up to the segment-level (or frame-level) captioning stage but consolidates multiple captions directly using an LLM. As shown in Tables 1, 2, and 3 of the main paper, SGVC significantly outperforms LLM summarization, indicating that scene graph consolidation effectively integrates information without introducing substantial errors.
>
> *  Artifact from the generation of segment-level (frame-level) captions:
> Analyzing the impact of segment-level (or frame-level) caption generation is more complex. However, converting video segments (or frames) into text is sufficiently effective for merging semantic information across multiple segments (or frames), facilitating more coherent video-level understanding. Note that the counterpart, maintaining visual representations for individual segments (or frames), is not interpretable and requires an additional module for consolidating the information from multiple segments (or frames).
>
>
> **2. Low performance of 'LLM summarization’ in Table 2**
>
> We appreciate the suggestions for additional references [1, 2, 3] to compare on ActivityNet Captions. However, we clarify that these methods are extensively trained on large-scale video-text datasets, including ActivityNet Captions, whereas our approach operates in a zero-shot setting without using target dataset annotations during training. Therefore, direct comparisons between [1, 2, 3] and SGVC are not meaningful.
>
> **3. Evaluation on YouCookII**
>
> We agree that evaluating on a dataset with longer videos, such as YouCookII [4], is meaningful. We are currently conducting experiments on YouCookII and expect that the benefits of SGVC will be particularly evident in datasets with long videos, similar to ActivityNet Captions. However, due to time constraints, we were unable to include these results in our initial response.
>
> [1] Huang, B. et al. VTimeLLM: Empower LLM to Grasp Video Moments. In CVPR 2024.
>
> [2] Tang, Y. et al. Empowering LLMs with Pseudo-Untrimmed Videos for Audio-Visual Temporal Understanding. In AAAI 2025.
>
> [3] Kim, M. et al. HiCM2^2 2: Hierarchical Compact Memory Modeling for Dense Video Captioning. In AAAI 2025.
>
> [4] Zhou, L. et al. Towards Automatic Learning of Procedures From Web Instructional Videos. In AAAI 2018.

---

> > ### Comment · Reviewer_38AV · 2025-04-07
> >
> > Thank you for your thorough rebuttal. I find your explanation regarding error propagation convincing, and I accept your clarification about comparisons with references [1,2,3] given your zero-shot approach.
> > However, I still believe evaluation on YouCookII with longer videos is necessary to fully demonstrate your method's capabilities. If possible, I recommend adding these experiments to the final version.
> > I'd like to maintain weak accept recommendation.

---

### Official Review · Reviewer_SHDF · 2025-03-13

**Overall Recommendation:** 1

**Summary:**

The authors propose a zero-shot video captioning approach that combines frame-level scene graphs from a video to obtain intermediate representations for caption generation. This method generates frame-level captions using an image VLM, converts them into scene graphs, and consolidates these graphs to produce comprehensive video-level descriptions. To achieve this, the authors leverage a lightweight graph-to-text model trained solely on text corpora, eliminating the need for video captioning annotations.

**Claims And Evidence:**

Yes

**Essential References Not Discussed:**

[1] Nguyen, Trong-Thuan, Pha Nguyen, and Khoa Luu. "Hig: Hierarchical interlacement graph approach to scene graph generation in video understanding." In Proceedings of the IEEE/CVF Conference on Computer Vision and Pattern Recognition, pp. 18384-18394. 2024.

**Experimental Designs Or Analyses:**

Yes

**Methods And Evaluation Criteria:**

Yes

**Other Comments Or Suggestions:**

See Weaknesses.

**Other Strengths And Weaknesses:**

Strengths:

The proposed approach for fine-grained video captioning through scene graph consolidation demonstrates significant strengths, including effective utilization of image-based vision-language models (VLMs) adapted successfully for video domains without requiring task-specific annotations or computationally intensive training.

Weaknesses:

1. Potential reliance on the quality and accuracy of frame-level captions generated by image VLMs, vulnerability to cumulative parsing errors in scene graphs.
2. Limited ability to capture subtle temporal dynamics between frames due to the discrete nature of the frame-level graph merging process.
3. Paper was written in a rush.

**Questions For Authors:**

See Weaknesses.

**Relation To Broader Scientific Literature:**

Need more comparisons with work [1].


[1] Nguyen, Trong-Thuan, Pha Nguyen, and Khoa Luu. "Hig: Hierarchical interlacement graph approach to scene graph generation in video understanding." In Proceedings of the IEEE/CVF Conference on Computer Vision and Pattern Recognition, pp. 18384-18394. 2024.

**Theoretical Claims:**

Yes

---

> ### Author Rebuttal · Authors · 2025-04-01
>
> **1. Reliance on the quality of frame-level captions and vulnerability to cumulative parsing errors in scene graphs**
>
> *  Error propagation is common in other algorithms:
> The primary objective of this work is to understand video content and generate fine-grained, video-level captions without encoding entire videos at once. This approach is especially effective for handling long videos, which require division into multiple segments or frames. However, consolidating information from these segments inevitably leads to cumulative errors—a challenge common to various algorithms despite their efforts to minimize it.
>
> *  Error quantification is difficult:
> Due to the zero-shot nature of our method, directly quantifying the cumulative error from abstracting segment-level (or frame-level) visual information into textual captions and extracting video-level representation based on a scene graph generation and consolidation is challenging. This is because existing methods employ fundamentally different algorithms, including variations in encoder and decoder choices for representation and captioning.
>
> *  An attempt to estimate the impact of cumulative parsing error:
> One way to isolate the parsing error in scene graph construction is to compare performance between LLM summarization and our approach, SGVC. Note that LLM summarization follows the same pipeline as SGVC up to the segment-level (or frame-level) captioning stage but consolidates multiple captions directly using an LLM. As shown in Tables 1, 2, and 3 of the main paper, SGVC significantly outperforms LLM summarization, indicating that scene graph consolidation effectively integrates information without introducing substantial errors.
>
> *  Artifact from the generation of segment-level (frame-level) captions:
> Analyzing the impact of segment-level (or frame-level) caption generation is more complex. However, converting video segments (or frames) into text is sufficiently effective for merging semantic information across multiple segments (or frames), facilitating more coherent video-level understanding. Note that the counterpart, maintaining visual representations for individual segments (or frames), is not interpretable and requires an additional module for consolidating the information from multiple segments (or frames).
>
>
> **2. Limited ability to capture subtle temporal dynamics between frames**
>
> As discussed in our response to #1, our core contribution lies in developing a pipeline for fine-grained video captioning without encoding entire videos at once, which involves scene graph generation and its consolidation. This allows for a variety of captioning models to be used; our approach is not restricted to frame-level captioners but can also incorporate video-level ones. Notably, the combination of a video-level encoder (InternVL2.5-1B [1]) and SGVC demonstrates promising performance, outperforming LLM summarization, although comparisons with external algorithms are not available at this time. For detailed results, please refer to our response to Reviewer 3YCD for #2.
>
>
>
>
>
>
>
>
> **3. Additional reference**
>
> We appreciate the reference to HIG [2] for video scene graph generation. This study offers a valuable technique for constructing more comprehensive scene graphs. However, our work is not a competitor to HIG; rather, it has significant potential to benefit from integrating HIG’s scene graph generation techniques to enhance performance. Additionally, HIG requires extensive training to learn a scene graph generation model, but since pretrained models are not publicly available, immediate integration into our approach may not be feasible.
>
>
> [1] Chen, Z. et al. Expanding Performance Boundaries of Open-Source Multimodal Models with Model, Data, and Test-Time Scaling. In arXiv, 2024.
>
> [2] Nguyen, T. et al. HIG: Hierarchical interlacement graph approach to scene graph generation in video understanding. In CVPR 2024.

---

### Decision · Program_Chairs · 2025-05-01

**Decision:**

Accept (poster)

**Comment:**

Paper proposes a video captioning approach that first extracts frame-level captions, parses them into intermediate scene graphs, consolidates those scene graphs and then uses this consolidates scene graph to produce a video-level caption without ever relying on video caption annotations. Paper received 1 x Reject, 1 x Weak Reject and 2 x Weak Accepts. Main concerns of reviewers centered around (1) fundamental limitations of the proposed approach, such as propagation of error and inability to observe and caption subtle actions that require understanding of motion [SHDF, 38AV], (2) lack of performance on certain benchmarks [38AV] or with respect to some SoTA methods [3YCD] (e.g., VideoPrism and VideoCoCa), (3) missing evaluation of inference-time cost [JfkZ], and (4) missing ablations with different VLMs [JfkZ]. Authors have provided rebuttals that successfully addressed many of the concerns, with an exception of (2). The issues brought up by [SHDF] and [38AV] regarding the fundamental limitations of the proposed approach are valid in AC's opinion. At the same time, AC also believes that proposed approach is not necessarily designed to serve as the ultimate solution for the video captioning, but rather as an intriguing benchmark that shows how much is possible with frame-based information extraction. In this context, paper makes an interesting and valuable contribution to the community. As such, AC believes the paper is valuable for the community and would make a good addition to ICML, thereby recommending ACCEPTANCE.